# DADM: Hallucination Detection in LLMs via Distance-Aware Distribution Modeling

## Abstract

Despite the remarkable advancements, large language models (LLMs) still frequently generate outputs that contain factually incorrect or contextually irrelevant information, commonly known as hallucinations. Detecting these hallucinations accurately and efficiently remains an open challenge, especially without relying on labeled datasets. Current methods primarily depend on internal activation or consistency of multiple responses for one prompt, limiting their effectiveness in capturing global semantic and distributional structures of truthful outputs. Besides, methods that estimate latent subspaces directly from mixed-quality data, suffer from noise contamination and imprecise geometric representations. To address these limitations, we propose a novel Distance-Aware Distribution Modeling (DADM) framework that operates in two stages: first, we apply an iterative distance-based process to select consistently truthful samples; second, we model the global distribution using normalizing flows, enabling accurate likelihood estimation by maximizing the likelihood of truthful samples and minimizing the likelihood of hallucinated samples. This two-stage design ensures both robust sample purification and expressive modeling of truthful generations, leading to interpretable confidence scores and more reliable hallucination detection. Extensive experiments on benchmark datasets demonstrate that our method consistently outperforms prior unsupervised approaches across multiple LLM settings.

## 1 Introduction

Large language models (LLMs) have demonstrated impressive capabilities in a variety of natural language processing tasks, including text summarization, translation, and question-answering (Zhao et al., 2023; Cao et al., 2023; Naveed et al., 2023). Despite their widespread success, a critical challenge remains: LLMs frequently generate responses containing factually incorrect or contextually irrelevant information, commonly referred to as hallucinations (Rawte et al., 2023; Bai et al., 2024; Huang et al., 2025). Such hallucinations significantly undermine the trustworthiness and applicability of LLMs, especially in domains like healthcare and scientific research.

However, manual annotation for hallucination detection in LLMs is labor-intensive and often impractical, particularly in rapidly evolving or specialized domains. To overcome this, recent work has explored unsupervised approaches that do not rely on explicit ground-truth labels. Some methods aim to detect inconsistencies between internal model signals and the generated output (Sriramanan et al., 2024). For example, MIND (Su et al., 2024) leverages internal activations of the model during generation to identify hallucinated content without requiring manual supervision. Besides, some research focuses on uncertainty estimation, where semantic entropy across multiple generations is used to flag outputs with high variability (Kuhn et al., 2023). Additionally, contrastive likelihood methods evaluate a model's sensitivity to prompt perturbations as a proxy for factual reliability (Burns et al., 2022). While these methods are scalable and do not require labeled data, they primarily operate at the level of individual examples. This limitation may lead to misclassification of rare but valid outputs or conflation of uncertainty with factual errors.

Another line of work is represented by HaloScope, which estimates hallucination likelihood by projecting LLM outputs onto a subspace constructed from unlabeled generations and measuring their distances to this subspace (Du et al., 2024). However, because the subspace is learned from all prompt-response pairs, including hallucinated samples, it provides a biased and noisy approximation

of the true semantic structure. The bias of the geometric representation subspace grows rapidly with the ratio of hallucinations, so the HaloScope may become degenerate and fail to accurately separate truthful and hallucinated responses, which hampers its reliability in practice.

To overcome the difficulties, we propose a novel two-stage Distance-Aware Distribution Modeling (DADM) framework for hallucination detection. Our key idea is to select a high-confidence truthful subset by utilizing the language model's latent representations and then learn the truthfulness distribution with this subset to distinguish a truthful response and a potentially hallucinated point. In contrast to prior approaches such as HaloScope that rely on noisy subspace estimates, our method leverages a clean subset of truthful responses, which remains stable in the presence of hallucinations.

To be specific, DADM firstly identifies a clean set of truthful examples based on a distance metric with robust estimations in Stage 1. This subset is obtained through an iterative refinement process, similar to prior approaches (Rousseeuw, 1984; Rousseeuw & Driessen, 1999), that progressively filters out likely hallucinations, resulting in a coherent set of high-confidence responses. Notably, our selection procedure is computationally efficient and remains robust as long as the proportion of hallucinated responses is below 50%. Leveraging this curated subset, we then capture the complex distribution of the feature space by training a normalizing flow model in Stage 2. Together, the two stages form a cohesive system: the initial selection provides reliable supervision for the flow model, and the learned likelihoods further refine detection.

Empirical results validate the effectiveness and robustness of DADM. Across a range of benchmark datasets spanning factual verification and commonsense reasoning, our DADM consistently outperforms existing unsupervised baselines. Specifically, we observe improvements of over 6% AUROC on factual tasks such as TriviaQA (Joshi et al., 2017) for OPT-6.7b model, and even larger gains exceeding 10% on more challenging reasoning benchmarks like CommonsenseQA (Talmor et al., 2019) for LLaMA-3.1-8b model. These results highlight the broad generalization capabilities of our method across different LLM architectures and task types, confirming its resilience even under ambiguous prompts and limited supervision.

In summary, we illustrate the full procedure in Figure 1 and highlight our main contributions:

- We propose a novel two-stage distance-aware distribution modeling framework. The central idea is to identify a clean subset of high-confidence truthful responses without relying on any labeled supervision by integrating an iterative distance-based process, which mitigates the effects of hallucinations and serves as a reliable foundation for confidence estimation.

- With the high-confidence truthful set, a distributional modeling based on normalizing flows is adapted to enable flexible and precise likelihood estimation over the hidden feature space and produce calibrated confidence scores to improve the hallucination detection.

- Our DADM framework offers strong practicality and flexibility for real-world applications. Extensive experiments across multiple language models and diverse datasets demonstrate the effectiveness and robustness of our method, achieving consistent improvements over state-of-the-art unsupervised baselines.

## 2 METHODOLOGY

### 2.1 PROBLEM SETUP

We formalize hallucination detection as a binary classification problem over prompt–response pairs produced by a fixed language model. Let $\mathcal{P}$ and $\mathcal{R}$ denote the spaces of natural-language prompts and model generations, respectively. We write $\Phi\colon \mathcal{P} \to \mathcal{R}$ for the fixed LLM under study, so that each prompt $\mathbf{x}_{\mathrm{p}} \in \mathcal{P}$ yields a response $\mathbf{x}_{\mathrm{g}} = \Phi(\mathbf{x}_{\mathrm{p}}) \in \mathcal{R}$. We therefore consider the joint space $\mathcal{X} = \mathcal{P} \times \mathcal{Q}$, where $(\mathbf{x}_{\mathrm{p}}, \mathbf{x}_{\mathrm{g}})$ is a prompt–response pair. Assume there exists an (unknown) "truthful" distribution $\mathcal{X}_{\mathrm{true}}$ over $\mathcal{X}$ and a "hallucinated" distribution $\mathcal{X}_{\mathrm{hal}}$ capturing ungrounded or fabricated outputs. Our goal is to learn a classifier such that

$$G(\mathbf{x}_{\mathrm{p}}, \mathbf{x}_{\mathrm{g}}) = \begin{cases} 1, & (\mathbf{x}_{\mathrm{p}}, \mathbf{x}_{\mathrm{g}}) \sim \mathcal{X}_{\mathrm{true}}, \\ 0, & (\mathbf{x}_{\mathrm{p}}, \mathbf{x}_{\mathrm{g}}) \sim \mathcal{X}_{\mathrm{hal}}, \end{cases}$$

which predicts whether a given pair is truthful (1) or hallucinated (0).

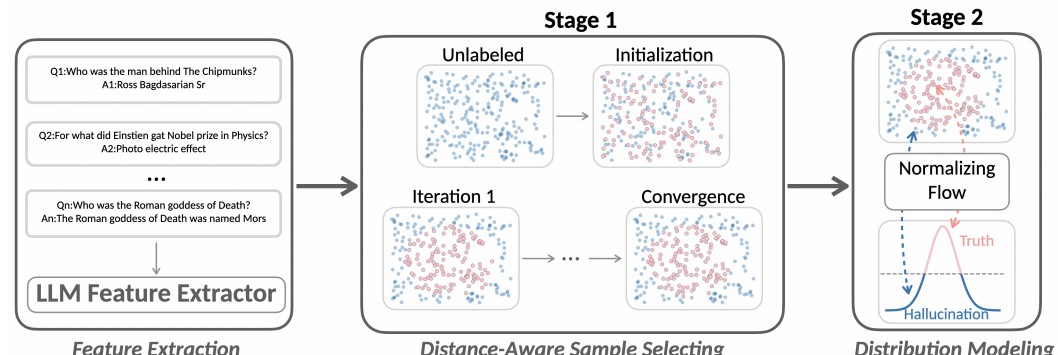

Figure 1: Overview of the DADM framework for hallucination detection. Given prompt-response pairs, the LLM feature extractor maps inputs into a feature space. In Stage 1, an iterative covariance-modified distance-based process selects a high-confidence subset of truthful responses. In Stage 2, a normalizing flow model is trained to model the underlying distribution and produce calibrated likelihood estimates for hallucination detection.

## 2.2 HIGH-CONFIDENCE SAMPLES VIA DISTANCE-BASED OUTLIER DETECTION

Let $\{(\mathbf{x}_i^{\mathrm{p}}, \mathbf{x}_i^{\mathrm{g}})\}_{i=1}^n$ be the unlabeled prompt–response pairs generated by one fixed LLM. We encode each prompt-response pair by feeding the entire pair as a prompt into the same LLM and extracting intermediate representations. Specifically, we denote the feature as $\mathbf{f}_i = \phi(\mathbf{x}_i^{\mathrm{p}}, \mathbf{x}_i^{\mathrm{g}}) \in \mathbb{R}^d$, where $\phi$ refers to the intermediate activation obtained from the LLM. Stacking these row-wise yields $\mathbf{F} = [\mathbf{f}_1^\top; \ldots; \mathbf{f}_n^\top] \in \mathbb{R}^{n \times d}$, and one can expect the feature space to reflect structural differences. It is quite natural to expect that truthful features will appear relatively similar and form tight clusters, since they are grounded in consistent factual content and share similar structural patterns. By contrast, hallucinated responses tend to be more arbitrary, diverse, and semantically inconsistent, which causes their representations to spread out more widely and lack a coherent structure.

This motivates treating hallucination detection as an outlier detection problem, where hallucinated responses correspond to potential outliers. Distance-based methods are widely used in high-dimensional outlier detection to identify samples that deviate from the data distribution in the embedding space (Knorr et al., 2000; Ro et al., 2015; Lee et al., 2018). These techniques are effective at capturing distributional deviations without requiring supervision. Inspired by these techniques, we develop an iterative framework that identifies a cohesive and self-consistent subset of feature indices.

Formally, we denote by $\mathcal{H} \subseteq [n]$ an index subset of cardinality $m$, where $[n] = \{1, 2, \ldots, n\}$ is the set of all sample indices. Define the center point as $\bar{\mathbf{c}}_{\mathcal{H}} \in \mathbb{R}^d$ of $\mathcal{H}$. The consistency of a sample $\mathbf{f}_i$ with respect to $\mathcal{H}$ is then quantified by its distance to the center $d(\mathbf{f}_i, \bar{\mathbf{c}}_{\mathcal{H}}) : \mathbb{R}^d \times \mathbb{R}^d \to \mathbb{R}^+$. Smaller values of $d(\mathbf{f}_i, \bar{\mathbf{c}}_{\mathcal{H}})$ indicate that $\mathbf{f}_i$ is more closely aligned with the center. The refinement procedure can be expressed as a two-step iterative update:

1. **Center Estimation.** Given the current index subset $\mathcal{H}_0 \subset [n]$, compute its center as

$$\bar{\mathbf{c}}_{\mathcal{H}_0} = \arg \min_{\mathbf{y} \in \mathbb{R}^d} \sum_{i \in \mathcal{H}_0} d(\mathbf{f}_i, \mathbf{y}). \tag{1}$$

2. **Subset Update.** With the center $\bar{\mathbf{c}}_{\mathcal{H}_0}$ fixed, update the subset by retaining the $m$ indices whose feature vectors are closest to the center:

$$\mathcal{H}_1 = \arg \min_{\substack{\mathcal{S} \subseteq [n] \\ |\mathcal{S}| = m}} \sum_{i \in \mathcal{S}} d(\mathbf{f}_i, \bar{\mathbf{c}}_{\mathcal{H}_0}). \tag{2}$$

Equivalently, this corresponds to ranking all samples by ascending distance to the center and selecting the top $m$ samples that are considered most truthful.

Starting from an initial random subset $\mathcal{H}_0$, the two steps are iterated until the index set stabilizes. The general framework only requires a distance function $d$, and the center estimation has explicit forms with some commonly-used choice of the distance metric. For example:

- If using Euclidean distance $d(\mathbf{f}_i, \mathbf{y}) = \|\mathbf{f}_i - \mathbf{y}\|_2^2$, the center estimation is the *arithmetic mean* $\bar{\mathbf{c}}_{\mathcal{H}_0} = |\mathcal{H}_0|^{-1} \sum_{i \in \mathcal{H}_0} \mathbf{f}_i$.
- If using Manhattan distance $d(\mathbf{f}_i, \mathbf{y}) = \|\mathbf{f}_i - \mathbf{y}\|_1$, the center estimation corresponds to the *coordinate-wise median*, which is widely used in robust statistics.

In our implementation, we consider a covariance-modified $L_2$ distance:

$$d(\mathbf{f}_i, (\bar{\mathbf{f}}_{\mathcal{H}_0}, \mathbf{D}_{\mathcal{H}_0})) = (\mathbf{f}_i - \bar{\mathbf{f}}_{\mathcal{H}_0})^\top \mathbf{D}_{\mathcal{H}_0}^{-1} (\mathbf{f}_i - \bar{\mathbf{f}}_{\mathcal{H}_0}), \tag{3}$$

where we consider $\bar{\mathbf{c}}_{\mathcal{H}_0} = (\bar{\mathbf{f}}_{\mathcal{H}_0}, \mathbf{D}_{\mathcal{H}_0})$: $\bar{\mathbf{f}}_{\mathcal{H}_0}$ is the robust estimator of mean and $\mathbf{D}_{\mathcal{H}_0}$ is the robust diagonal covariance matrix. The scale-invariance of this distance ensures that all feature dimensions contribute fairly. Here, the diagonal covariance is employed instead of the classical covariance matrix since the robust estimation of the covariance matrix is unreliable when the sample size $n$ is on the order of the feature dimension $d$ as discussed in (Ro et al., 2015). The diagonal assumption could also reduce both storage and computational complexity from $O(d^2)$ to $O(d)$.

To select the high-confidence truthful set, we adopt a trimmed refinement procedure in a similar spirit to least-trimmed-squares approaches (Rousseeuw, 1984; Rousseeuw & Driessen, 1999). We start from an initial random subset of size $m$, iteratively re-estimate its local mean and diagonal covariance, and retain the $m$ responses closest to this center. The full procedure is summarized as follows:

1. **Initialization.** Randomly select an initial truthful index set $\mathcal{H}_0 \subset [n]$ of cardinality $m$.
2. **Center Estimation.** Based on formula (1) and distance (3), we first compute the sample mean $\bar{\mathbf{f}}_{\mathcal{H}_0}$ and a simplified diagonal covariance matrix $\mathbf{D}_{\mathcal{H}_0}$ from the current index subset $\mathcal{H}_0$. We omit the factor in the covariance estimation for simplicity:

$$\bar{\mathbf{f}}_{\mathcal{H}_0} = \frac{1}{m} \sum_{i \in \mathcal{H}_0} \mathbf{f}_i, \qquad \mathbf{D}_{\mathcal{H}_0} = \mathrm{diag}\Big(\frac{1}{m} \sum_{i \in \mathcal{H}_0} (\mathbf{f}_i - \bar{\mathbf{f}}_{\mathcal{H}_0})(\mathbf{f}_i - \bar{\mathbf{f}}_{\mathcal{H}_0})^\top\Big).$$

3. **Subset Update.** Form the refined index subset using formula (2) and distance (3):

$$\mathcal{H}_1 = \big\{\pi(1), \pi(2), \ldots, \pi(m)\big\},$$

where $\pi(1), \pi(2), \ldots, \pi(n)$ denote the corresponding indices after ranking.
4. **Convergence Check.** Replace $\mathcal{H}_0 \leftarrow \mathcal{H}_1$ and repeat Steps 2–3 until either $\mathcal{H}_1 = \mathcal{H}_0$ (no change) or a maximum of $T$ iterations is reached.
5. **Selection Across Different Initialization.** After completing $K$ independent runs of Steps 1–4 with different initialization, each yielding a core $\mathcal{H}_0$ and its total cumulative distances, choose the set with the smallest cumulative distances as the final high-confidence set $\mathcal{H}_0^{\text{best}}$.

Since choosing the subset size $m$ involves a trade-off, smaller values (around $0.1n$) tend to produce a cleaner, safer core but risk missing some truthful samples, whereas larger $m$ may include hallucinations. Therefore, we carefully select $m$ within the range between $0.1n$ to $0.5n$ to balance reliability and coverage. We validate this choice of $m$ in Section 3.3. To mitigate sensitivity to local optima, we perform multiple random initializations in Step 5, each independently exploring different regions of the solution space. The configuration with the minimal cumulative distance is selected as the final high-confidence set, providing a robust foundation for downstream hallucination detection.

The following theorem ensures that the cumulative distance of the iterative procedure monotonically decreases and converges to a local minimum.

**Theorem 1.** *Let $\mathcal{H}_0 \subset [n]$ be an index subset of size $m$. Compute the covariance-modified $L_2$ distances $d(\mathbf{f}_i, (\bar{\mathbf{f}}_{\mathcal{H}_0}, \mathbf{D}_{\mathcal{H}_0}))$ for all $i \in [n]$, where $\bar{\mathbf{f}}_{\mathcal{H}_0}$ and $\mathbf{D}_{\mathcal{H}_0}$ are the mean and diagonal covariance estimated from $\mathcal{H}_0$. Let $\mathcal{H}_1$ be the set of indices corresponding to the $m$ smallest distances, i.e.,*

$$\mathcal{H}_1 = \{\pi(1), \pi(2), \ldots, \pi(m)\},$$

*where $\pi$ is the ordering of indices such that*

$$d(\mathbf{f}_{\pi(1)}, (\bar{\mathbf{f}}_{\mathcal{H}_0}, \mathbf{D}_{\mathcal{H}_0})) \leq d(\mathbf{f}_{\pi(2)}, (\bar{\mathbf{f}}_{\mathcal{H}_0}, \mathbf{D}_{\mathcal{H}_0})) \leq \cdots \leq d(\mathbf{f}_{\pi(n)}, (\bar{\mathbf{f}}_{\mathcal{H}_0}, \mathbf{D}_{\mathcal{H}_0})).$$

*Then, computing the mean and covariance based on $\mathcal{H}_1$, the cumulative distance satisfies*

$$\sum_{i \in \mathcal{H}_1} d(\mathbf{f}_i, (\bar{\mathbf{f}}_{\mathcal{H}_1}, \mathbf{D}_{\mathcal{H}_1})) \leq \sum_{i \in \mathcal{H}_0} d(\mathbf{f}_i, (\bar{\mathbf{f}}_{\mathcal{H}_0}, \mathbf{D}_{\mathcal{H}_0})),$$

*with equality if and only if $\mathcal{H}_1 = \mathcal{H}_0$.*

## 2.3 Distribution Modeling for Hallucination Detection

In Stage 1, we identify a tightly clustered core $\mathcal{H}_0^{\text{best}} \subset [n]$ serving as a high-confidence set of non-hallucinated examples. This stage plays a crucial role in providing a reliable anchor for hallucination detection, offering a principled geometric approach under minimal supervision. While the initial distance-based selection tends to be safe, it may miss certain truthful examples that lie further from the estimated core. Moreover, the method models feature distributions only accounting for mean and diagonal covariance, which may not fully capture more complex structures present in real-world data.

To address these limitations while building upon the strengths of Stage 1, we introduce Stage 2 that further refines the modeling of the feature space. The key idea is to employ distribution learning algorithms to model the distribution of truthful responses on the refined $\mathcal{H}_0^{\text{best}}$. Specifically, normalizing flows achieve precise likelihood estimation and flexible distribution transformation by stacking invertible coupling layers, enabling accurate modeling of complex data distributions for improved uncertainty quantification (Kobyzev et al., 2020; Papamakarios et al., 2021). Therefore, we train a normalizing flow on $\mathcal{H}_0^{\text{best}}$ to flexibly learn the underlying complex distribution and to provide calibrated likelihood estimates for individual samples.

**Training Objective.** We denote the normalizing flow model as $g_\psi$, which transforms input feature vectors $\mathbf{f} \in \mathcal{F}$ into latent representations $\mathbf{z} = g_\psi(\mathbf{f})$, where a standard Gaussian distribution is defined. The log likelihood is computed via the change of variables formula:

$$\log p_{\mathcal{F}}(\mathbf{f}) = \log p_{\mathcal{Z}}(g_\psi(\mathbf{f})) + \log \left| \det \left( \frac{\partial g_\psi(\mathbf{f})}{\partial \mathbf{f}} \right) \right|.$$

In practice, we implement $g_\psi$ using a stack of invertible $1 \times 1$ convolutions, affine coupling layers, and activation normalization, following the Glow architecture Kingma & Dhariwal (2018). To emphasize the distinction between hallucinated and truthful samples, the model is trained using a dual-objective loss based on the selected truthful features $\mathcal{H}_0^{\text{best}}$ and its complement $[n] \setminus \mathcal{H}_0^{\text{best}}$ (Zhao et al., 2025):

- **Truthful Objective:** We aim to maximize the log-likelihood over the set $\mathcal{H}_0^{\text{best}}$, thereby encouraging the model to faithfully capture the distribution of truthful features.
- **Hallucination Suppression:** To mitigate hallucination, we penalize the likelihood assigned to elements in $[n] \setminus \mathcal{H}_0^{\text{best}}$ by minimizing the softplus transformation of their log-likelihoods. The use of the softplus function ensures optimization stability and mitigates numerical instability when likelihood values are close to zero or exceedingly small.

The total loss is defined as:

$$\mathcal{L} = \mathcal{L}_{\text{truth}} + \lambda \cdot \mathcal{L}_{\text{hallu}}$$
$$= -\frac{1}{m} \sum_{i \in \mathcal{H}_0^{best}} \log p_{\mathcal{F}}(\mathbf{f}_i) + \frac{\lambda}{n-m} \sum_{j \in [n] \setminus \mathcal{H}_0^{best}} \log \left( 1 + p_{\mathcal{F}}(\mathbf{f}_j) \right),$$

where $m$ and $n - m$ are the number of truthful and hallucinated samples respectively, and $\lambda$ is a hyperparameter that balances the two objectives. This distribution modeling allows for a smoother and more adaptive decision boundary, which is particularly beneficial for hallucination detection.

**Scoring and Detection.** Together, the two stages complement each other: the first determines approximately dense and reliable samples, while the second provides fine-grained, continuous likelihood estimates across the feature space. Once trained, the normalizing flow model assigns a log likelihood score to each feature vector $\mathbf{f}$, providing a continuous measure of confidence for hallucination detection. A high log likelihood indicates that the sample aligns well with the learned distribution of truthful samples, while a low score suggests potential hallucination. We define the scoring function as $p_{\mathcal{F}}(\mathbf{f})$, and use it to construct a binary classifier $c(\mathbf{f})$ that labels a feature as truthful if its score exceeds a threshold $\tau$. Formally,

$$c(\mathbf{f}) = \begin{cases} 1, & \text{if } p_{\mathcal{F}}(\mathbf{f}) > \tau \\ 0, & \text{if } p_{\mathcal{F}}(\mathbf{f}) \leq \tau \end{cases},$$

where $c(\mathbf{f}) = 1$ denotes a truthful prediction and $c(\mathbf{f}) = 0$ indicates hallucination. For practical deployment, we select the threshold $\tau$ based on a criterion, such as the false positive rate (5%). For comparison with experiments, we directly compute the AUROC from the continuous scores.

## 3 EXPERIMENTS

### 3.1 EXPERIMENTAL SETUP

To rigorously evaluate the performance of our hallucination detection framework DADM, we perform experiments across four diverse benchmark datasets. TruthfulQA (Lin et al., 2021) (817 validation examples) focuses on open-domain QA in conversational contexts. For closed-book question answering, we use a deduplicated validation split of the TriviaQA (rc.nocontext subset) (Joshi et al., 2017), containing 9,960 factoid-style questions. To assess performance in knowledge comprehension, we include the SciQ validation set (Johannes Welbl, 2017) (1,000 examples) and the CommonsenseQA validation set (Talmor et al., 2019) (1,221 examples), involving-choice questions designed to test domain-specific and commonsense reasoning. Each dataset is partitioned into three subsets: a 75% unlabeled train set used for distance-aware distribution learning, a 25% held-out test set for evaluation, and 100-example validation set (randomly sampled from the training split). The template for response generation is as follows: {Answer the question concisely.  Q: {question} A:}

We apply our method to two instruction-tuned foundation models: OPT-6.7b (Zhang et al., 2022) and LLaMA-3.1-8b (Grattafiori et al., 2024). These models are selected for their strong open-ended generation capabilities and widespread adoption in downstream applications, making them representative for hallucination detection. Features are obtained from the outputs of each Transformer blocks, allowing us to analyze the effect of different layers on detection performance. To assess the effectiveness of our hallucination detection methods, we adopt the area under the receiver operating characteristic curve (AUROC) as our primary evaluation metric. Following Du et al. (2024), we use BLUERT (Sellam et al., 2020), a learned metric to label the groud-truth of hallucinations. A prompt-response pair is labeled as truthful if its BLUERT score exceeds a threshold of 0.5.

For comprehensive evaluation, we select seven hallucination detection methods that represent three paradigms: (1) probability-based metrics (Perplexity (Ren et al., 2022), Maximum Sequence Probability (MSP) (Fadeeva et al., 2023)) analyzing prediction confidence, (2) distance measures (FisherRao Distance (Darrin et al., 2022), Haloscope (Du et al., 2024)) quantifying hidden state variations, and (3) semantic consistency checks (Lexical Similarity (Fomicheva et al., 2020), Semantic Entropy (Kuhn et al., 2023), EigenScore (Chen et al., 2024)) evaluating conceptual alignment. This selection strategically covers both model-intrinsic signals (probability distributions, hidden states) and model-extrinsic validations (lexical/semantic coherence), enabling a comprehensive comparative analysis of diverse hallucination detection approaches. We reproduce baseline methods using the LM-Polygraph framework to ensure a fair and consistent comparison (Fadeeva et al., 2023).

**Implementation Details**   For Stage 1, we perform $K = 10$ random reinitializations. The subset ratio $m/n$ and the feature extraction layer are tuned according to validation performance. Empirically, $m/n$ is selected within the range $[0.1, 0.5]$, and the feature layer is chosen from layers 5 to 16. We employ a normalizing flow architecture based on Glow (Kingma & Dhariwal, 2018), the model processes an input consisting of a $4,096$ dimensional feature vector through a series of 8 coupling layers. We set Adam (Kingma & Ba, 2014) optimizer, with hyperparameter set as follows: a learning rate of $2 \times 10^{-4}$, batch size of 24, $\beta_1 = 0.9$, $\beta_2 = 0.999$, $\epsilon = 10^{-8}$, and weight decay of $10^{-5}$. We train the normalizing flow model for a maximum of 20 epochs. Additionally, we select $\lambda = 1$ due to superior empirical performance. All experiments are implemented on a single NVIDIA Tesla A100 GPU with 80GB of memory. Each experiment was repeated with three different random seeds, and the results are reported as the mean with standard deviation.

### 3.2 RESULTS

The results in Table 1 clearly demonstrate the superiority of our method across different datasets and LLMs. On OPT-6.7b, our method achieves the best AUROC scores on all four benchmarks, significantly surpassing the best baseline HaloScope, by 9.93, 2.74, 5.05, and 4.67 points respectively. Particularly on challenging commonsense reasoning tasks like CommonsenseQA and SciQ, our method achieves over 79 and 80 AUROC, respectively, highlighting its robustness beyond surface-level fact checking. Similarly, on the LLaMA-3.1-8b model, our method again establishes new state-of-the-art performance, achieving 71.58 on TruthfulQA, 77.06 on TriviaQA, 78.30 on CommonsenseQA, and 80.24 on SciQ. Compared to other methods, our method maintains stable performance

gains, particularly in knowledge-intensive tasks. These substantial margins, especially on complex datasets requiring deeper reasoning, underline the broad effectiveness of our detection method.

Overall, these results confirm that our framework not only outperforms prior single-output hallucination detectors across both open domain (TriviaQA, TruthfulQA) and knowledge comprehension (CommonsenseQA, SciQ) tasks, but also generalizes well across model scales and question types. The strong and consistent gains suggest that our method can serve as a robust and versatile solution for hallucination detection in diverse real world applications.

Table 1: Performance comparison of hallucination detection methods across OPT-6.7b and LLaMA-3.1-8b models on four datasets. Our distance-aware distribution modeling framework achieves state-of-the-art results on all datasets. Our experiment was conducted three times with different random seeds, and the results are presented as the mean and standard deviation.

| Model | Method | TruthfulQA | TriviaQA | CommonSenseQA | SciQ |
|---|---|---|---|---|---|
| OPT-6.7b | Perplexity | 57.41 | 60.49 | 57.21 | 60.75 |
| | Fisher Rao | 52.76 | 60.48 | 60.38 | 59.23 |
| | Lexical Similarity | 53.97 | 61.09 | 63.32 | 57.94 |
| | MSP | 50.11 | 58.33 | 52.05 | 58.24 |
| | Semantic Entropy | 50.69 | 64.80 | 55.33 | 58.87 |
| | EigenScore | 50.97 | 61.42 | 51.00 | 64.62 |
| | HaloScope | $69.64_{\pm5.13}$ | $62.94_{\pm3.55}$ | $74.28_{\pm2.68}$ | $76.01_{\pm2.37}$ |
| | DADM (Ours) | $\mathbf{79.57}_{\pm1.10}$ | $\mathbf{65.68}_{\pm1.02}$ | $\mathbf{79.33}_{\pm0.42}$ | $\mathbf{80.68}_{\pm1.83}$ |
| LLaMA-3.1-8b | Perplexity | 61.43 | 76.32 | 50.60 | 64.51 |
| | Fisher Rao | 56.85 | 67.14 | 59.13 | 55.29 |
| | Lexical Similarity | 59.45 | 66.19 | 50.85 | 60.84 |
| | MSP | 57.39 | 67.29 | 62.91 | 68.91 |
| | Semantic Entropy | 62.48 | 72.77 | 61.32 | 65.97 |
| | EigenScore | 52.49 | 70.52 | 68.32 | 66.10 |
| | HaloScope | $70.27_{\pm0.92}$ | $73.51_{\pm0.28}$ | $64.66_{\pm0.56}$ | $76.01_{\pm0.03}$ |
| | DADM (Ours) | $\mathbf{71.58}_{\pm0.13}$ | $\mathbf{77.06}_{\pm0.53}$ | $\mathbf{78.30}_{\pm0.41}$ | $\mathbf{80.24}_{\pm0.82}$ |

## 3.3 ABLATION STUDIES

We conduct a series of ablation studies on all datasets to systematically evaluate the key components and design choices of our distance-aware distribution modeling hallucination detection framework. Specifically, we analyze (i) the effect of the retained truthful feature size ($m$) in Stage 1, (ii) the impact of feature extraction layer selection, (iii) the importance of employing a normalizing flow architecture for flexible distribution modeling, and (iv) the scalability of our method to larger language models. These studies provide detailed insights into how each component contributes to the overall effectiveness and robustness of the framework.

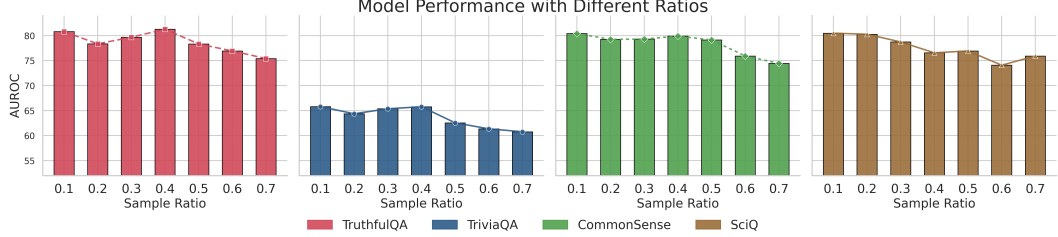

Figure 2: Effect of truthful sample ratio ($m/n$) on AUROC performance for OPT-6.7b across different datasets. We vary $m/n$ from 0.1 to 0.7 in the initial stage 1, and the best performance is typically achieved when $m/n$ lies between 0.1 and 0.5.

**Effect of Truthful Features Ratio ($m/n$).** Stage 1 plays a critical role in isolating a reliable core of truthful responses from unlabeled generations. To assess the impact of the truthful samples ratio, we vary $m/n$ from 0.1 to 0.7, corresponding to retaining 10% to 70% samples as truthful samples. For

each setting, we conduct the full iterative distance-based refinement followed by likelihood estimation with normalizing flows and evaluate performance using AUROC on the test set.

Our analysis reveals a consistent non-monotonic pattern across all datasets (Figure 2) on OPT-6.7b model. The AUROC scores initially improve with increasing sample ratios, peaking between 0.1 and 0.5 (TruthfulQA achieves 81.2 at 0.4, TriviaQA achieves 65.5 at 0.4 and CommonSenseQA achieves 80.4 at 0.1), followed by a marked performance degradation beyond a 0.5 ratio. This phenomenon can be attributed to potential hallucination contamination in higher sample ratios, as evidenced by the steepest decline in TriviaQA and CommonSenseQA from 0.5 to 0.7. This trend suggests that larger sample ratios (beyond 0.5) may introduce noisy or hallucinated examples, degrading model reliability. These findings empirically support the importance of identifying a clean set of high-confidence truthful responses. Such a clean subset serves as the proxy for the subsequent distribution modeling stage. Consequently, choosing a smaller sample ratio ($\leq 0.5$) offers a safer and more robust configuration for hallucination detection.

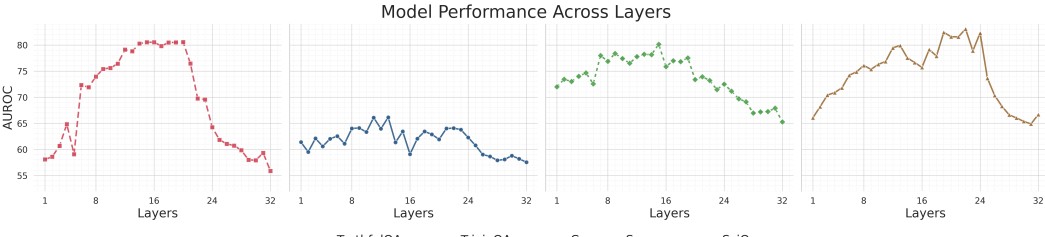

Figure 3: AUROC performance across different feature extraction layers of OPT-6.7b. Te best performance is typically achieved when features are extracted from intermediate layers.

**Effect of Layer Feature.** We investigate how the choice of feature extraction layer influences hallucination detection performance. For each dataset, we compute AUROC scores by independently applying the distance-aware high-confidence sample selection and normalizing flow pipeline with the fixed optimal truthful sample ratio. We report the results on OPT-6.7b across various datasets in Figure 3, and observe that the detection performance varies substantially across layers. Across all datasets, AUROC scores typically improve from the earliest layers, reach a peak between layers 8 and 24, and then gradually degrade toward deeper layers. For instance, on CommonSenseQA, AUROC peaks above 80.4 at layer 15, whereas in other datasets, the best performance is also observed around middle layers. In contrast, features from very early layers and very late layers consistently yield lower AUROC scores, often dropping by more than 10 points compared to the optimal middle-layer performance. Overall, these results suggest that intermediate layer representations are more effective at capturing rich contextual dependencies that are critical for distinguishing hallucinations from truthful contents. In contrast, early layers primarily encode shallow patterns, while deeper layers tend to become overly specialized toward next-token prediction, potentially obscuring the finer distinctions between truthful and hallucinated responses.

**Importance of Normalizing Flow Architecture.** A key component of our approach is the use of normalizing flows for distribution modeling. We compare three modeling settings: (i) directly using Stage 1, where a class center is learned and scoring is based on the covariance-modified distance to the center; (ii) adding a 2-layer MLP with ReLU activation, and the first layer maps to 1024 dimension; and (iii) incorporating a normalizing flow to model the selected feature distribution. As shown in Figure 4, adding normalizing flows in Stage 2 brings consistent improvements over the stage 1 results, and outperforms the linear probing baseline across different settings. For OPT-6.7b, results based on Stage 1 provide the baseline performance, while linear probing leads to only slight gains in AUROC scores across datasets. The normalizing flow further elevates results by 0.69-7.68 points, achieving state-of-the-art values. On LLaMA-3.1-8b, AUROC scores based on Stage 1 start with 63.85, 77.38, 76.76, and 78.95, while linear probing contributes moderate improvements of up to 3.78 points. The normalizing flow again outperforms baselines, adding further gains ranging from 1.88 to 6.96 points. These results demonstrate that while results in Stage 1 provide initial improvements, the flexible modeling power of the normalizing flow delivers further substantial enhancements.

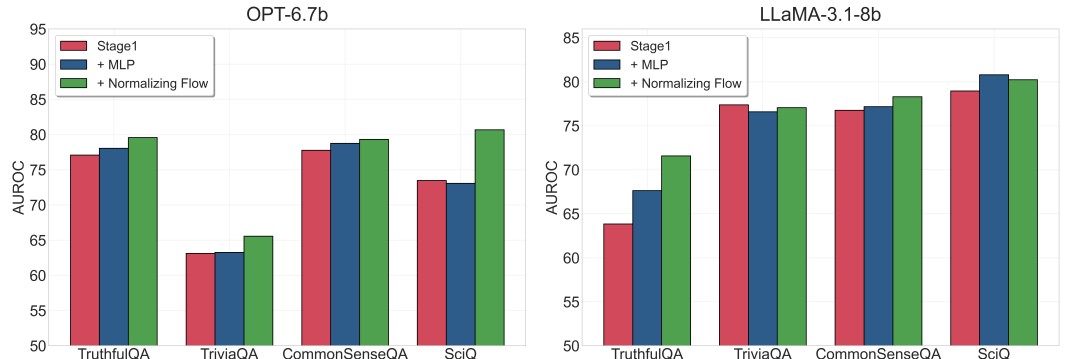

Figure 4: Comparison of hallucination detection AUROC on OPT-6.7b and LLaMA-3.1-8b across four benchmarks, highlighting the performance gains achieved by incorporating normalizing flow.

**Generalizability and Long-form QA Evaluation.** To evaluate the generalization capability of DADM beyond the datasets used during training, we further test the model on unseen out-of-distribution datasets. We evaluate the generalizability in Figure 5, DADM maintains strong performance across different source–target pairs. We also extend our analysis to long-form QA tasks, which require models to generate multi-sentence and context-rich responses. Specifically, we evaluate on the CLAPNQ dataset (Rosenthal et al., 2025), using the first 1,000 annotated examples from its 1,954 annotated examples in the training dataset. The task is structured as follows: {Answer the question based on the information in the given passage: Passage: {passage}, Q: {question}, A:}. We utilize Qwen3-4b as the language model and employ DeepSeek-V3.2 as the LLM-judge metric. As shown in Figure 6, DADM achieves competitive or superior performance compared to baseline methods. DADM achieves the highest score of 64.34, surpassing all other methods. These results demonstrate that the model effectively captures transferable hallucination representations and generalizes well to unseen datasets and complex generative scenarios in long-form QA settings.

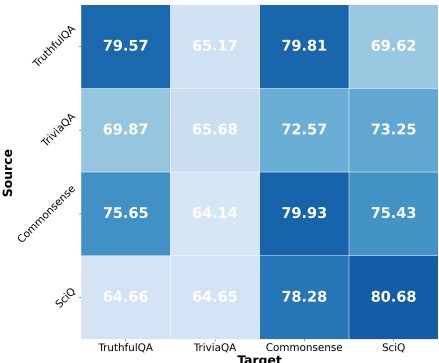

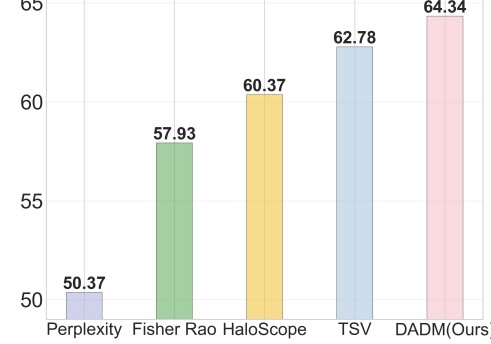

Figure 5: Generalization Results.

Figure 6: Results on long-form QA.

## 4 CONCLUSION

In this work, we proposed a novel distance-aware distribution modeling (DADM) framework for hallucination detection in large language models. By leveraging distance-aware high-confidence sample selection via iterative Mahalanobis-distance refinement, followed by expressive probabilistic modeling using normalizing flows, our method effectively detect hallucinations without relying on labeled data. Extensive experiments across multiple datasets and LLM models demonstrate that our approach consistently outperforms state-of-the-art unsupervised baselines. Our results highlight the importance of modeling global feature distribution properties rather than relying solely on token-level uncertainty or local activation patterns. The framework is flexible, interpretable, and scalable, offering a practical solution for hallucination detection in real world LLM deployments.

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

## A  PSEUDOCODE

In this section, we present the detailed pseudocode of the proposed Distance-Aware Sample Selecting algorithm in Algorithm 1. This stage aims to select a representative subset of $m$ samples from the full dataset by minimizing the cumulative distance to the subset mean, under a diagonal covariance assumption. The method performs $K$ random restarts, each with up to $T$ inner refinement steps, to ensure robustness and convergence toward high-quality selections.

## B  PROOF OF THEOREM 1

**Theorem 1.** *Let $\mathcal{H}_0 \subset [n]$ be an index subset of size $m$. Compute the covariance-modified $L_2$ distances $d(\mathbf{f}_i, (\bar{\mathbf{f}}_{\mathcal{H}_0}, \mathbf{D}_{\mathcal{H}_0}))$ for all $i \in [n]$, where $\bar{\mathbf{f}}_{\mathcal{H}_0}$ and $\mathbf{D}_{\mathcal{H}_0}$ are the mean and diagonal covariance estimated from $\mathcal{H}_0$. Let $\mathcal{H}_1$ be the set of indices corresponding to the $m$ smallest distances, i.e.,*

$$\mathcal{H}_1 = \{\pi(1), \pi(2), \ldots, \pi(m)\},$$

*where $\pi$ is the ordering of indices such that*

$$d(\mathbf{f}_{\pi(1)}, (\bar{\mathbf{f}}_{\mathcal{H}_0}, \mathbf{D}_{\mathcal{H}_0})) \le d(\mathbf{f}_{\pi(2)}, (\bar{\mathbf{f}}_{\mathcal{H}_0}, \mathbf{D}_{\mathcal{H}_0})) \le \cdots \le d(\mathbf{f}_{\pi(n)}, (\bar{\mathbf{f}}_{\mathcal{H}_0}, \mathbf{D}_{\mathcal{H}_0})).$$

*Then, computing the mean and covariance based on $\mathcal{H}_1$, the cumulative distance satisfies*

$$\sum_{i \in \mathcal{H}_1} d(\mathbf{f}_i, (\bar{\mathbf{f}}_{\mathcal{H}_1}, \mathbf{D}_{\mathcal{H}_1})) \le \sum_{i \in \mathcal{H}_0} d(\mathbf{f}_i, (\bar{\mathbf{f}}_{\mathcal{H}_0}, \mathbf{D}_{\mathcal{H}_0})),$$

*with equality if and only if $\mathcal{H}_1 = \mathcal{H}_0$.*

The proof relies on the following lemma.

**Lemma 1.** *Let $\mathcal{H} \subset [n]$ be a fixed size-$m$ subset, and define $J(\bar{\mathbf{f}}, \mathbf{D} \mid \mathcal{H}) := \sum_{i \in \mathcal{H}} d(\mathbf{f}_i, (\bar{\mathbf{f}}, \mathbf{D}))$. Among all diagonal positive-definite matrices $\mathbf{D} \succ 0$ and all $\bar{\mathbf{f}} \in \mathbb{R}^d$, the pair $(\bar{\mathbf{f}}_{\mathcal{H}}, \mathbf{D}_{\mathcal{H}})$ defined in Algorithm 1 minimizes $J(\bar{\mathbf{f}}, \mathbf{D} \mid \mathcal{H})$.*

*Proof.* Let $\mathbf{D} = \text{diag}(\sigma_1^2, \ldots, \sigma_d^2)$, $\bar{\mathbf{f}} = (\mu_1, \ldots, \mu_d)^\top$, and $\mathbf{f}_{ij}$ denote the $j$-th component of $\mathbf{f}_i$. The objective becomes:

$$J(\bar{\mathbf{f}}, \mathbf{D} \mid \mathcal{H}) = \sum_{j=1}^{d} \frac{1}{\sigma_j^2} \sum_{i \in \mathcal{H}} (\mathbf{f}_{ij} - \mu_j)^2.$$

**Algorithm 1:** Distance-Aware Sample Selecting

**Input:** $\mathcal{F}$: full feature set of $n$ samples;
$m$: subset size;
$K$: number of random restarts;
$T$: maximal inner iterations.
**Output:** $\mathcal{H}_0^{\text{best}}$: size-$m$ index set.

1   $\mathcal{H}_0^{\text{best}} \leftarrow \emptyset$;
2   $min\_cum\_distances \leftarrow \infty$;          `// initial best objective`
3   **for** $k = 1$ **to** $K$ **do**
4     **Randomly initialize** $\mathcal{H}_0 \subset [n]$ with $m$ samples;
5     **for** $t = 1$ **to** $T_{\max}$ **do**                `// inner refinement`
6       $\bar{\mathbf{f}}_{\mathcal{H}_0} = \dfrac{1}{h} \sum\limits_{i \in \mathcal{H}_0} \mathbf{f}_i; \mathbf{D}_{\mathcal{H}_0} = \text{diag}\left( \dfrac{1}{h} \sum\limits_{i \in \mathcal{H}_0} (\mathbf{f}_i - \bar{\mathbf{f}}_{\mathcal{H}_0})(\mathbf{f}_i - \bar{\mathbf{f}}_{\mathcal{H}_0})^\top \right)$;
7       **for** $i = 1$ **to** $n$ **do**
8         $d(\mathbf{f}_i, (\bar{\mathbf{f}}_{\mathcal{H}_0}, \mathbf{D}_{\mathcal{H}_0})) \leftarrow (\mathbf{f}_i - \bar{\mathbf{f}}_{\mathcal{H}_0})^\top \mathbf{D}_{\mathcal{H}_0}^{-1} (\mathbf{f}_i - \bar{\mathbf{f}}_{\mathcal{H}_0})$;
9       $\mathcal{H}_1 \leftarrow$ indices of $m$ samples with the smallest $d(\mathbf{f}_i, (\bar{\mathbf{f}}_{\mathcal{H}_0}, \mathbf{D}_{\mathcal{H}_0}))$;
10      **if** $\mathcal{H}_1 = \mathcal{H}_0$ **then break**;
11      **else**
12         $\mathcal{H}_0 \leftarrow \mathcal{H}_1$
13     $cum\_distances \leftarrow \sum\limits_{i \in \mathcal{H}_0} d(\mathbf{f}_i, (\bar{\mathbf{f}}_{\mathcal{H}_0}, \mathbf{D}_{\mathcal{H}_0}))$;
14     **if** $cum\_distances < min\_cum\_distances$ **then**
15       $\mathcal{H}_0^{\text{best}} \leftarrow \mathcal{H}_0$;
16       $min\_cum\_distances \leftarrow cum\_distances$;
17   **return** $\mathcal{H}_0^{best}$

The optimization decouples across dimensions due to diagonal $\mathbf{D}$.
**Step 1: Optimal Mean.** For fixed $\sigma_j^2$, set the derivative with respect to $\mu_j$ to zero:

$$\frac{\partial}{\partial \mu_j} \sum_{i \in \mathcal{H}} (\mathbf{f}_{ij} - \mu_j)^2 = -2 \sum_{i \in \mathcal{H}} (\mathbf{f}_{ij} - \mu_j) = 0 \quad \implies \quad \mu_j = \frac{1}{m} \sum_{i \in \mathcal{H}} \mathbf{f}_{ij} = [\bar{\mathbf{f}}_{\mathcal{H}}]_j.$$

**Step 2: Optimal Variance.** For each given $j$ , we consider:

$$g_j(\sigma_j^2) := \frac{1}{\sigma_j^2} \sum_{i \in \mathcal{H}} (\mathbf{f}_{ij} - \mu_j)^2.$$

Compute the second derivative and we have:

$$\frac{\partial^2 g_j}{\partial (\sigma_j^2)^2} = \frac{2}{(\sigma_j^2)^3} \sum_{i \in \mathcal{H}} (\mathbf{f}_{ij} - \mu_j)^2 > 0.$$

The unique minimum occurs at:

$$\sigma_j^2 = \frac{1}{m} \sum_{i \in \mathcal{H}} (\mathbf{f}_{ij} - \mu_j)^2 = [\mathbf{D}_{\mathcal{H}}]_{jj}.$$

Thus, $(\bar{\mathbf{f}}_{\mathcal{H}}, \mathbf{D}_{\mathcal{H}})$ attains the minimum. $\qquad\qquad\qquad\qquad\qquad\qquad\qquad\qquad\qquad\quad$ □

*Proof of Theorem 1.* Compute distances using $(\bar{\mathbf{f}}_{\mathcal{H}_0}, \mathbf{D}_{\mathcal{H}_0})$, then select the $m$ smallest distances to form $\mathcal{H}_1$ and we have:

$$\sum_{i \in \mathcal{H}_1} d(\mathbf{f}_i, (\bar{\mathbf{f}}_{\mathcal{H}_0}, \mathbf{D}_{\mathcal{H}_0})) \leq \sum_{i \in \mathcal{H}_0} d(\mathbf{f}_i, (\bar{\mathbf{f}}_{\mathcal{H}_0}, \mathbf{D}_{\mathcal{H}_0})). \tag{4}$$

Next, applying Lemma 1 to $\mathcal{H} = \mathcal{H}_1$ yields

$$\sum_{i \in \mathcal{H}_1} d(\mathbf{f}_i, (\bar{\mathbf{f}}_{\mathcal{H}_1}, \mathbf{D}_{\mathcal{H}_1})) \leq \sum_{i \in \mathcal{H}_1} d(\mathbf{f}_i, (\bar{\mathbf{f}}_{\mathcal{H}_0}, \mathbf{D}_{\mathcal{H}_0})).$$

Combining this with (4) gives $\sum_{i \in \mathcal{H}_1} d(\mathbf{f}_i, (\bar{\mathbf{f}}_{\mathcal{H}_1}, \mathbf{D}_{\mathcal{H}_1})) \leq \sum_{i \in \mathcal{H}_0} d(\mathbf{f}_i, (\bar{\mathbf{f}}_{\mathcal{H}_0}, \mathbf{D}_{\mathcal{H}_0}))$. □

## C VISUALIZATION OF FEATURE DISTRIBUTION

To gain deeper insights into the geometric characteristics of the features extracted by the LLM, we apply t-SNE to visualize the feature embeddings across four datasets in Figure 7. In each plot, truthful samples are shown in red and hallucinated samples in blue. The visualizations are based on projected features into two dimensions using t-SNE. Across all datasets, we observe a consistent trend: truthful samples tend to form compact and coherent clusters, while hallucinated samples appear more scattered. This property supports the core intuition behind Stage 1 of our method: truthful generations tend to occupy dense regions of the embedding space, whereas hallucinated outputs can be treated as outliers.

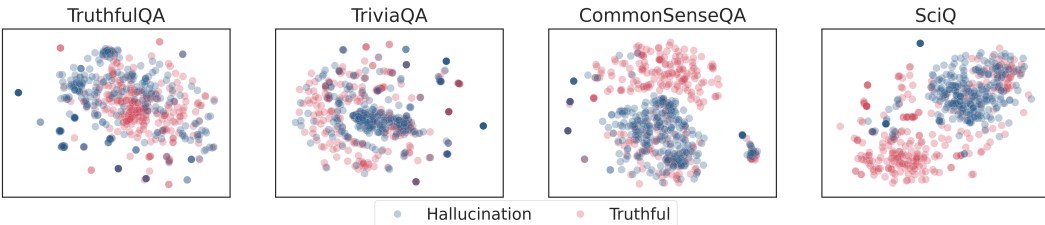

Figure 7: T-SNE visualization of feature distributions extracted by OPT-6.7b across different datasets. Truthful samples (red) form compact clusters, while hallucinated samples (blue) are more dispersed, illustrating the geometric intuition behind Stage 1 of our method.

Besides, we perform a t-SNE visualization of the selected truthful features, comprising approximately 10% of the total samples, based on features extracted from LLaMA-3-8.1b and OPT-6.7b on the TruthfulQA and TriviaQA datasets (Figure 8). These visualizations correspond to the output of Stage 1 in our framework. The results show that our method effectively identifies predominantly truthful samples and substantially reduces the proportion of hallucinations. Notably, the samples selected from LLaMA-3 exhibit a clearer separation between truthful and hallucinated samples, reflecting the stronger discriminative capacity of its representations. While the selected samples are largely composed of truthful samples, a small number of hallucinated samples still remain. This observation motivates Stage 2 of our approach, where we further improve the results by modeling the truthful distribution with a more flexible normalizing flow model.

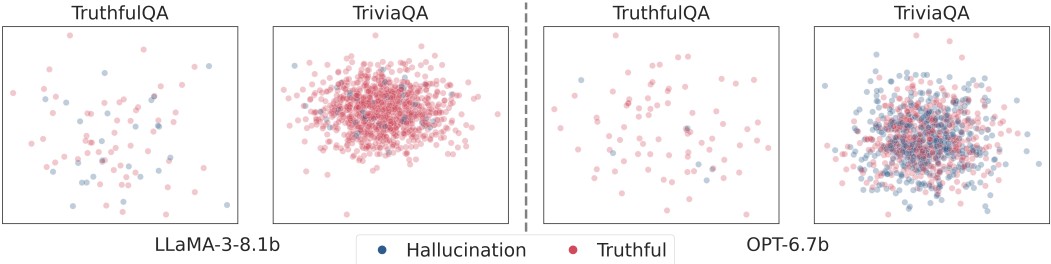

Figure 8: T-SNE visualization of subsets selected using Algorithm 1 (10% of total samples) from representations generated by LLaMA-3-8.1b (left) and OPT-6.7b (right). The selection algorithm effectively prioritizes truthful samples, especially when applied to stronger LLaMA-3-8.1b model.

## D    COMPUTATIONAL TIME

We provide a detailed comparison of the computational time of our method DADM against several key baselines: HaloScope, EigenScore and Semantic Entropy. DADM and HaloScope are training-based models that require a training phase, whereas EigenScore and Semantic Entropy do not involve a separate training stage but instead compute scores based on pre-trained model features.

We benchmarked the total end-to-end wall-clock time for each method, where for DADM and HaloScope, the time includes feature extraction, training, and inference. For EigenScore and Semantic Entropy, the time includes feature extraction and the scoring process. All experiments were conducted on a single NVIDIA H100 GPU.

Table 2: End-to-end wall-clock time for different methods.

| Method | Requires Training | Total Time (hour) |
|---|---|---|
| HaloScope | Yes | 0.28 |
| DADM (Ours) | Yes | 0.57 |
| EigenScore | No | 1.04 |
| Semantic Entropy | No | 1.09 |

The results of this comparison are summarized in Table 2. As shown in the table, DADM is significantly faster than both EigenScore and Semantic Entropy, which do not require training and rely on pre-trained features. It is slightly slower than HaloScope, another training-based method. Moreover, the computational time of DADM can be further reduced by decreasing the dimensionality of the intermediate layers within the normalizing flows.

## E    COMPARISON WITH TSV

To ensure a more comprehensive comparison, we further include the TSV (Park et al., 2025) for LLaMA-3.1-8b and Qwen3-4b (Yang et al., 2025). As summarized in Table 3 and Table 4, TSV exhibits strong performance across various datasets, validating its strength as a competitive baseline. However, our DADM method consistently surpasses or matches TSV, achieving notable gains on CommonSenseQA and SciQ, which highlights the robustness and general superiority of our approach across diverse evaluation scenarios.

Table 3: Performance comparison of hallucination detection methods across LLaMA-3.1-8b models on four datasets.

| Model | Method | TruthfulQA | TriviaQA | CommonSenseQA | SciQ |
|---|---|---|---|---|---|
| | HaloScope | $70.27_{\pm 0.92}$ | $73.51_{\pm 0.28}$ | $64.66_{\pm 0.56}$ | $76.01_{\pm 0.03}$ |
| LLaMA-3.1-8b | TSV | $\mathbf{71.78}_{\pm 0.34}$ | $\mathbf{78.59}_{\pm 0.72}$ | $67.83_{\pm 2.18}$ | $70.40_{\pm 1.84}$ |
| | DADM (Ours) | $71.58_{\pm 0.13}$ | $77.06_{\pm 0.53}$ | $\mathbf{78.30}_{\pm 0.41}$ | $\mathbf{80.24}_{\pm 0.82}$ |

Table 4: Performance comparison of hallucination detection methods across Qwen3-4b model on four datasets. DADM consistently outperforms both HaloScope and TSV across all datasets.

| Model | Method | TruthfulQA | TriviaQA | CommonSenseQA | SciQ |
|---|---|---|---|---|---|
| | HaloScope | 72.81 | 61.46 | 60.66 | 61.81 |
| Qwen3-4b | TSV | 64.64 | 67.14 | 68.47 | 66.96 |
| | DADM | **75.74** | **69.34** | **70.63** | **67.34** |

## F    ORACLE EXPERIMENTS

To further examine the impact of truthful supervision in Stage 2 modeling, we introduce an oracle analysis using all BLEURT-true samples for Stage 2 training.

Specifically, we consider two settings: (a) **Partial Truthful Subset:** Only use BLEURT-true samples as the truthful subset for Stage 2 modeling, without treating the complementary samples as negatives. (b) **Full Truthful Subset with Negatives:** Use all BLEURT-true samples as the truthful subset, and treating the complement as negative samples.

These two setups analyze the influence of truthful-sample coverage and negative-sample supervision on final performance. The results in Table 5 show that incorporating penalty signals for negative samples (W. Penalty) leads to higher oracle performance, highlighting the importance of incorporating untruthful supervision. In contrast, using only truthful samples without penalties (W.O. Penalty) results in a noticeable drop, indicating the necessity of contrastive learning signals. Our DADM method achieves competitive results, demonstrating its robustness even without explicit oracle penalties.

Table 5: Oracle performance comparison of hallucination detection methods across OPT-6.7b model on four datasets.

| Model | Method | TruthfulQA | TriviaQA | CommonSenseQA | SciQ |
|---|---|---|---|---|---|
| | W. Penalty | **85.43** | **71.62** | **81.91** | 76.37 |
| OPT-6.7b | W.O. Penalty | 80.15 | 63.92 | 79.63 | 75.14 |
| | DADM | 79.57 | 65.68 | 79.33 | **80.68** |

## G EVALUATION WITH OTHER LABELING METRICS

Our training procedure is entirely unsupervised, and BLEURT is only employed as an automatic labeling tool to annotate hallucination examples for evaluation. To ensure a more comprehensive and reliable evaluation, we further assess our method under two alternative metrics: ROUGE-L evaluation (Lin, 2004) and the DeepSeek-V3.2 (Liu et al., 2024) LLM-as-a-judge evaluation, which has been shown to align better with human preferences.

The ROUGE-L metric emphasizes sequence-based similarity, while the DeepSeek evaluation utilizes large language models for contextual judgment. We use DeepSeek-V3.2 to respond with 'Yes' if the generated answer matches any of the gold standard answers, indicating it is truthful. If the generated answer does not match any gold standard answers, the model responds with 'No' and we label the response as a hallucination. As shown in Table 6, our DADM method maintains consistently strong performance across both evaluation protocols, confirming its robustness and stability under different labeling schemes. These results validate that the effectiveness of DADM is not dependent on a specific metric, but generalizes well across diverse evaluation metrics.

## H MORE ABLATION RESULTS

**Effect of Penalty on Complementary Samples.** We investigate the impact of treating the complement of the truthful subset as negative samples during the Stage 2 modeling process. Our design may introduce mild label noise by penalizing a few truthful responses while emphasizes true positive detection. When the penalty on complementary samples is removed (W.O. Penalty), performance drops consistently across different datasets in Table 7. This shows incorporating penalties for the complementary set strengthens the discriminative capacity of the model and improves overall hallucination detection performance.

**Averaged Multi-layer Features.** In the current implementation of DADM, features are extracted from individual layers rather than aggregating all layers. To assess whether feature aggregation benefits performance, we tested a more tractable variant that averages hidden representations across all layers. As reported in Table 8, the averaged representation performs notably worse than the layer-wise approach, confirming that selecting a specific hidden layer provides a more effective representation for hallucination detection.

**Larger Models.** To further assess the scalability and robustness of our approach, we conduct experiments using two larger models, LLaMA-2-13b and OPT-13b (Zhang et al., 2022), which

Table 6: Performance comparison of hallucination detection methods across OPT-6.7b with ROUGE and DeepSeek labeling.

| Metric | Method | TruthfulQA | TriviaQA | CommonSenseQA | SciQ |
|--------|--------|-----------|----------|---------------|------|
| ROUGE-L | Perplexity | 55.99 | 50.23 | 60.94 | 51.29 |
| | Fisher Rao | 52.43 | 51.38 | 57.61 | 59.92 |
| | Lexical Similarity | 50.16 | 51.38 | 68.39 | 51.08 |
| | MSP | 53.08 | 52.76 | 52.71 | 50.29 |
| | Semantic Entropy | 53.42 | 56.54 | 64.61 | 53.04 |
| | EigenScore | 54.76 | 50.91 | 55.67 | 65.33 |
| | HaloScope | 81.08 | 84.66 | **90.54** | 80.24 |
| | DADM (Ours) | **85.26** | **86.11** | 84.19 | **85.19** |
| DeepSeek | Perplexity | 53.38 | 50.67 | 52.30 | 54.83 |
| | Fisher Rao | 56.32 | 51.25 | 56.98 | 54.62 |
| | Lexical Similarity | 55.72 | 51.77 | 50.92 | 50.58 |
| | MSP | 58.91 | 52.15 | 56.04 | 57.40 |
| | Semantic Entropy | 52.21 | 54.24 | 52.11 | 51.58 |
| | EigenScore | 54.29 | 51.47 | 52.92 | 51.00 |
| | HaloScope | 63.31 | 58.45 | 51.13 | 59.02 |
| | DADM (Ours) | **67.62** | **60.28** | **63.76** | **62.16** |

Table 7: Performance comparison of hallucination detection methods across OPT-6.7b model on four datasets.

| Model | Method | TruthfulQA | TriviaQA | CommonSenseQA | SciQ |
|-------|--------|-----------|----------|---------------|------|
| OPT-6.7b | W.O. Penalty | 73.30 | 63.81 | 78.95 | 76.41 |
| | DADM | **79.57** | **65.68** | **79.33** | **80.68** |

offer significantly greater capacity compared to OPT-6.7b and LLaMA-3.1-8b. As shown in Table 9, our method consistently outperforms the HaloScope baseline across all datasets and models. On TruthfulQA, our approach achieves AUROC scores of 85.73 and 89.98 for LLaMA-2-13b and OPT-13b, representing improvements of 5.36 and 7.57, respectively. On TriviaQA, we observe even larger gains, with an AUROC of 86.08 on LLaMA-2-13b (18.4-point improvement) and a 3.71-point improvement on OPT-13b, suggesting our method particularly excels at detecting factual hallucinations in knowledge-intensive tasks. Consistent advantages are also observed on CommonSenseQA and SciQ datasets, where DADM achieves better performance over the baseline across both model architectures. These results indicate that as model size increases, our method is able to more effectively leverage the additional capacity to distinguish between hallucinated and truthful outputs. The improvements across architectures and datasets further demonstrate that our approach generalizes well beyond specific model families, highlighting its robustness in more powerful LLMs.

# I LIMITATION

Our proposed DADM framework demonstrates strong empirical performance but has some limitations. In the first stage, it relies on selecting a high-quality subset of truthful samples through an iterative covariance-modified distance-based filtering process, retaining only 10%–30% of the unlabeled data.

Table 8: AUROC comparison between layer-wise features and averaged multi-layer features in DADM.

| Model | Method | TruthfulQA | TriviaQA | CommonSenseQA | SciQ |
|-------|--------|-----------|----------|---------------|------|
| OPT-6.7b | Average | 66.93 | 57.91 | 62.05 | 69.94 |
| | Layer-wise | **79.57** | **65.68** | **79.33** | **80.68** |

Table 9: Performance comparison between HaloScope and our method on larger models (LLaMA-2-13b and OPT-13b), demonstrating that our method consistently outperforms HaloScope.

| Model | Method | TruthfulQA | TriviaQA | CommonSenseQA | SciQ |
|-------|--------|------------|----------|---------------|------|
| LLaMA-2-13b | HaloScope | 80.37 | 67.68 | 72.66 | 79.80 |
|  | DADM (Ours) | **85.73** | **86.08** | **83.24** | **84.51** |
| OPT-13b | HaloScope | 82.41 | 59.66 | 77.13 | 74.34 |
|  | DADM (Ours) | **89.98** | **63.37** | **79.51** | **78.78** |

This conservative approach may underrepresent the true data distribution, potentially weakening generalization in the second stage. Additionally, using BLEURT to label hallucinations may introduce inaccuracies into the ground truth. Finally, although the normalizing flows used in the second stage offer flexibility in modeling complex distributions, we currently evaluate them primarily in comparison to the initial sample selection and a simple linear probing baseline. Exploring alternative density estimators or classification strategies could improve performance and is a promising direction for future research.

## J    BROADER IMPACTS

The proposed hallucination detection method has the potential to significantly improve the reliability and trustworthiness of LLMs in real world by identifying misleading or fabricated content. This advancement could be especially impactful in domains such as education, healthcare, and scientific communication. However, caution is necessary when applying this method to specialized contexts, as some hallucinations may still go undetected.

## K    THE USE OF LARGE LANGUAGE MODELS

We clearly describe the usage of large language model as a core component of the proposed hallucination detection method, detailing how they are integrated and utilized within the research. We also used the LLM to assist in improving the coherence of the manuscript.

