# OpenReview forum: "DADM: Hallucination Detection in LLMs via Distance-Aware Distribution Modeling"
_ICLR.cc/2026/Conference — Submitted to ICLR 2026_

### Official Review · Reviewer_5E77 · 2025-10-15

**Soundness:** 2
**Presentation:** 3
**Contribution:** 3
**Rating:** 4
**Confidence:** 4

**Summary:**

This paper proposed a new approach to detect hallucination with unlabeled data. The approach consists of two stages: high-confidence sample selection and flow model training. The authors first iteratively estimated the center of a set of representations and selected top-m highly confident truthful samples based on a distance metric. Then, the authors trained a flow model to predict whether a representation of a generation is truthful or not. The authors conducted experiments on four datasets with two LLMs, showing that their proposed approach outperforms other baselines. The authors also conducted ablation studies to justify their design choices.

**Strengths:**

1. The writing is clear and easy to follow.
2. The proposed method does not require labeled data, which would be easier to deploy in the real world.
3. The authors conducted extensive experiments to justify their design choices.

**Weaknesses:**

1. **Basic assumption.** The proposed method based on one assumption: "hallucinated responses tend to be more arbitrary, diverse, and semantically inconsistent, which causes their representations to spread out more widely and lack a coherent structure" (Line 135-136). Is there any reference to support this claim? As I can imagine that hallucinations can be fluent and plausible, like saying an event happened in 2021 while actually happening in 2022. In such cases, the hallucinated response may still have a highly semantically similar representation to factual content. Actually, in [1], the author has shown that the representation of truthful and hallucinated generations is highly overlapped, questioning the soundness of this paper.
2. **Baseline and LLMs.** A strong baseline, TSV [1], is overlooked. The authors may want to include it in the experiment. In addition, OPT-6.7b is relatively outdated. The authors could conduct experiments on some newer LLMs like Qwen3, to make the result more convincing.
3. **Generalizability.** The proposed method finds the high-confidence samples and trains a flow model on a single dataset. It is unclear how the model can generalize to out-of-distribution data. The author could report the cross-dataset performance to demonstrate the generalizability.

[1]: Steer LLM Latents for Hallucination Detection (2025)

**Questions:**

1. Line 106: $X_{hallu}\to X_{hal}$.
2. The current experiments were mainly conducted on short-form QA. I wonder whether DADM can be applied to long-form QA, which is a more realistic setting.

---

> ### Author Response · Authors · 2025-11-21
> **Response to Reviewer 5E77**
>
> We sincerely appreciate the reviewer's thoughtful and constructive feedback. Your comments have helped us clarify key assumption, new baselines, and demonstrate the generalizability of the work. In the following, we respond to each point in detail.
>
> **W1**. Basic assumption may not hold.
> **A1**: We appreciate the reviewer’s concern regarding the underlying assumption. Our claim is that truthful samples tend to form concentrated clusters in appropriate representation layers, whereas hallucinations exhibit more dispersed patterns. Empirically, as shown in Fig. 5, features from truthful responses consistently display compact clustering, while hallucinated responses are far more scattered. This observation motivates our Stage 1 design, which extracts a high-confidence truthful set.
>
> We would also like to clarify that the representation overlap between truthful and hallucinated responses reported in [1] (Fig. 1) is based on the final-layer features. Our ablation studies also indicate that the final layer features do not perform well in distinguishing hallucinated content from truthful content. In contrast, intermediate layers tend to provide much better separation between truthful and hallucinated responses. This insight is crucial because it aligns with the core of our method, which relies on the feature representations from intermediate layers to identify hallucinations effectively.
>
> **W2**. Baseline and LLMs are overlooked.
> **A2**: We appreciate the reviewer’s suggestion to include TSV as a baseline in our experiments. We are reproducing TSV and  will include it in future work to provide a more comprehensive comparison. We also conduct additional experiments on the newer LLM Qwen3 to further validate the effectiveness of DADM. As shown in Table 1, DADM outperforms Haloscope on Qwen3. Additional comparison results will be incorporated into the manuscript in the revision.
>
> ```plain
> Table 1. AUROC (%) of normalizing flows on Qwen3 model.
> ```
>
> | qwen3-4b  | TruthfulQA | TriviaQA | CommonSenseQA | SciQ  |
> | --------- | ---------- | -------- | ------------- | ----- |
> | Haloscope | 72.81      | 61.46    | 60.66         | 61.81 |
> | DAMD      | 75.74      | 69.34    | 70.63         | 67.34 |
>
>
> **W3**. Report the cross-dataset performance to demonstrate the generalizability.
> **A3**: Thank you for pointing out the need for clarity regarding the generalizability of our method. We now present the cross-dataset AUROC performance across the four QA datasets in Table 2. to demonstrate the generalizability. Models trained on one dataset consistently maintain competitive performance when transferred to the remaining datasets. We will include these results in the revised version of the manuscript.
>
> ```plain
> Table 2. Cross-dataset generalizability results across four QA datasets.
> ```
>
> | Source \ Target| **TruthfulQA** | **TriviaQA** | **Commonsense** | **SciQ** |
> | --------------------------------------------------- | -------------- | ------------ | --------------- | -------- |
> | **TruthfulQA**                                      | 79.57          | 65.17        | 79.81           | 69.62    |
> | **TriviaQA**                                        | 69.87          | 65.68        | 72.57           | 73.25    |
> | **Commonsense**                                     | 75.65          | 64.14        | 79.93           | 75.43    |
> | **SciQ**                                            | 64.66          | 64.65        | 78.28           | 80.68    |
>
> **Q**. The current experiments were mainly conducted on short-form QA. I wonder whether DADM can be applied to long-form QA, which is a more realistic setting.
> **A**: We appreciate the reviewer’s question regarding the applicability of DADM to long-form QA. At this stage, we are still identifying suitable long-form QA datasets that are appropriate for hallucination detection. Once an appropriate benchmark is selected and the experiments are completed, we will update the manuscript accordingly. Thank you for the insightful suggestion!
>
> [1] Park, Seongheon; Du, Xuefeng; Yeh, Min-Hsuan; Wang, Haobo; Li, Yixuan. Steer LLM Latents for Hallucination Detection. ICML 2025

---

> ### Comment · Reviewer_5E77 · 2025-11-21
>
> I thank the authors for their responses. The experimental results mainly addressed my concern. However, for **W1**, Figure 5 does not provide a good justification in my perspective. In that figure, the cluster of hallucinated data in Trivia QA, CommonSense QA, and SciQ are actually more compact than the cluster of truthful data. In addition, there is a significant overlap between the representations of hallucination and truthful data in Figure 5, which also aligns with the finding of the TSV paper. I expect the authors to further validate their claim.
>
> I will keep my score for now, but I may increase the score depending on the follow-up discussion and the revision.

---

> > ### Author Response · Authors · 2025-12-04
> > **Response to Reviewer 5E77 (Follow up)**
> >
> > We sincerely thank the reviewer for the response. Here are our explanations concerning W1, W2 and Q1.
> >
> > **W1**. Basic assumption may not hold.
> > **A1**: We appreciate the reviewer’s responses concerning Figure 5. Our intention is to identify a compact subset of samples that consistently form dense regions in the feature space. As shown in Figure 5, these  truthful samples (red points) exhibit more stable and coherent representations, forming a noticeably tight cluster. Besides, we have now included additional experiments on a new long-form QA dataset CLAPNQ, and we observe the same clustering paradigm.
> >
> > Since our approach is fully unsupervised, it necessarily relies on certain assumptions about the feature space. We believe our assumptions are reasonable and empirically supported, but we are also actively exploring strategies for scenarios where the assumptions may not hold. We thank the reviewer for this insightful suggestion.
> >
> > **W2**. Baseline and LLMs are overlooked.
> >
> > **A2**: Thanks for your suggestions. To ensure a more comprehensive comparison, we further include the TSV for LLaMA-3.1-8b and Qwen3-4b in Table 1 and Table2, our DADM method consistently surpasses or matches TSV, achieving notable gains on CommonSenseQA and SciQ, which highlights the robustness and general superiority of our approach across diverse evaluation scenarios.
> >
> > ```plain
> > Table 1.  Performance comparison of hallucination detection methods across LLaMA-3.1-8b model on four datasets.
> > ```
> >
> > | **Model**    | **Method**      | **TruthfulQA** | **TriviaQA** | **CommonSenseQA** | **SciQ**  |
> > | ------------ | --------------- | -------------- | ------------ | ----------------- | --------- |
> > | LLaMA-3.1-8b | **HaloScope**   | 70.27          | 73.51        | 64.66             | 76.01     |
> > |              | **TSV**         | **71.78**      | **78.59**    | 67.83             | 70.40     |
> > |              | **DADM (Ours)** | 71.58          | 77.06        | **78.30**         | **80.24** |
> >
> >
> > ```plain
> > Table 2.  Performance comparison of hallucination detection methods across Qwen3-4b model on four datasets.
> > ```
> >
> > | **Model**    | **Method**      | **TruthfulQA** | **TriviaQA** | **CommonSenseQA** | **SciQ**  |
> > | ------------ | --------------- | -------------- | ------------ | ----------------- | --------- |
> > | LLaMA-3.1-8b | **HaloScope**   | 72.81          | 61.46        | 60.66             | 61.81     |
> > |              | **TSV**         | 64.64          | 67.14        | 68.47             | 66.96     |
> > |              | **DADM (Ours)** | **75.74**      | **69.34**    | **70.63**         | **67.34** |
> >
> >
> > **Q1**: The current experiments were mainly conducted on short-form QA. I wonder whether DADM can be applied to long-form QA, which is a more realistic setting.
> >
> > **A1**: Thanks for your valuable suggestions. We now add a long-form QA task on CLAPNQ dataset. We utilize Qwen3-4b as the language model and employ DeepSeek-V3.2 as the LLM-judge metric. DADM achieves competitive or superior performance compared to baseline methods. DADM achieves the highest score of 64.34, surpassing all other methods.
> >
> > ```plain
> > Table 3. AUROC (%) of different methods evaluated on LLaMA-2-7b using the TruthfulQA dataset.
> > ```
> >
> > | **Model**    | **Method**      | **CLAPNQ** |
> > | ------------ | --------------- | ---------- |
> > | **Qwen3-4b** | **Perplexity**  | 50.37      |
> > |              | **Fisher Rao**  | 57.93      |
> > |              | **HaloScope**   | 60.37      |
> > |              | **TSV**         | 62.78      |
> > |              | **DADM (Ours)** | 64.34      |

---

### Official Review · Reviewer_uhnD · 2025-10-20

**Soundness:** 2
**Presentation:** 3
**Contribution:** 3
**Rating:** 4
**Confidence:** 3

**Summary:**

This paper proposes a two-stage unsupervised framework called DADM (Distance-Aware Distribution Modeling) for detecting hallucinations in large language models (LLMs). The authors argue that existing unsupervised methods, such as HaloScope, which directly construct a representation subspace from noisy data containing hallucinations, suffer from performance degradation due to data contamination.
To address this issue, the DADM framework operates in two stages:
Stage 1: Distance-aware sample selection.
This stage applies an iterative, distance-based outlier detection process to automatically select a high-confidence core subset of truthful samples from unlabeled LLM-generated outputs.
Stage 2: Distribution modeling.
After obtaining the purified truthful subset, this stage employs Normalizing Flows to model the feature distribution of the subset, learning a precise confidence score by maximizing the likelihood of truthful samples while minimizing that of potential hallucinations.
The main contribution of this paper lies in its innovative two-stage “purification–modeling” design, which mitigates the noise contamination problem in existing approaches by first identifying a clean data subset, leading to more robust hallucination detection. Experiments on multiple benchmark datasets and LLMs demonstrate that DADM outperforms existing unsupervised methods.

**Strengths:**

Clear problem orientation:
The paper accurately identifies the core weakness of existing unsupervised hallucination detection methods—particularly HaloScope—in handling noisy data, and proposes a well-targeted solution.

Extensive experimental coverage:
The study conducts comprehensive experiments across multiple models (e.g., OPT, LLaMA series) and benchmarks spanning factual QA and commonsense reasoning, accompanied by detailed ablation studies that demonstrate the method’s effectiveness.

Significant performance improvement:
Experimental results show that DADM consistently and substantially outperforms all baseline methods across all test scenarios, with particularly notable gains in the AUROC metric.

**Weaknesses:**

Implicit dependence on a labeled validation set:
The method requires a labeled validation set to tune key hyperparameters (the retained sample ratio (m/n) and the number of feature extraction layers), which contradicts the paper’s claim of being “fully unsupervised.” The authors should more candidly discuss this prerequisite and its limitation on the method’s practicality.

Questionable validity of the core assumption:
The approach relies on the assumption that truthful samples cluster together while hallucinated samples are scattered. This assumption may not hold in tasks requiring creativity or diverse responses, thereby constraining the applicability of the method.

High hyperparameter sensitivity:
As shown in the ablation studies (Fig. 2 and Fig. 3), the model’s performance is highly sensitive to the (m/n) ratio and the number of feature extraction layers. Improper settings can cause sharp performance degradation, resulting in high tuning costs in practical deployment—contradicting the goal of providing a robust solution.

**Questions:**

On the dependence on the validation set:
How do the authors view the issue that, in practical scenarios (e.g., in a completely new domain), a labeled validation set may be unavailable for tuning (m/n) and the number of feature extraction layers? Is there a heuristic, label-free approach to set these key parameters?

On the applicability of the core assumption:
In what types of tasks or data distributions do you think DADM might underperform? For instance, if a task’s correct answers are inherently diverse (multi-modal), making truthful samples fail to form a single compact cluster, would DADM’s first stage mistakenly exclude some truthful samples as outliers?

On the reliability of the evaluation:
Given that the BLEURT-based labels used for evaluation are themselves noisy, how do you think this noise might affect the two stages of DADM? In particular, during the first stage, if some hallucinated samples are incorrectly labeled as “truthful” by BLEURT, could they contaminate the selected high-confidence core subset?

---

> ### Author Response · Authors · 2025-11-21
> **Response to Reviewer uhnD**
>
> We thank the reviewer for the valuable feedback. These feedback will help enhance the practicality of DADM in real-world scenarios. We address each of the points you raised in detail below.
>
> **W1 & Q1**. Implicit dependence on a labeled validation set
> **A1**: We appreciate the reviewer’s observation regarding the use of a labeled validation set. While we do use a labeled validation set to tune key hyperparameters such as the ratio and the number of feature extraction layers, it is important to clarify that these are not supervisory parameters in the sense of providing labels for training the main model. These are simply hyperparameters that affect model performance. Similar procedures are used in prior work: [1] selects the SVD decomposition rank $k$ and the score threshold using a validation set, while [2] performs hyperparameter search using validation data. As demonstrated in our ablation studies in Fig 2, the performance of DADM is quite stable with respect to different hyper-parameter $m/n$ ratio. If a labeled validation set is unavailable, our empirical ablation results suggest that setting the ratio to around 0.3 and using intermediate-layer features provide reasonable default choices in practice.
>
> **W2 & Q2**. Questionable validity of the core assumption
> **A2**: Thank you for raising this point. To clarify, our approach does not assume that all truthful responses form a single compact cluster. Rather, the method is designed to identify and focus on the most confident cluster. If multiple valid answer clusters are possible, our method aims to identify and focus on the most confident cluster of truthful responses, prioritizing reliability for more accurate hallucination detection.
>
> **W3.** High hyperparameter sensitivity
> **A3**:  We agree with the reviewer’s observation that the performance of our method is sensitive to hyper-parameters such as the $m/n$ ratio and the number of feature extraction layers. However, we would like to clarify the following: In our ablation studies (Fig. 2), we found that when the $m/n$ ratio is within approximately 0.1 to 0.5, performance remains stable and high across datasets. Similarly, for the number of feature extraction layers, we show that middle-layers (Fig. 3) consistently yield better outcomes (which aligns with trends seen in [1] [2]). We are also exploring feature aggregation across layers (averaging features from multiple layers) to further reduce sensitivity. However, this comes with the trade-off of potentially incorporating “weaker” layers that may  degrade overall performance. As shown in Table 1, using features from a specific layer yields better performance than aggregated features.
>
> ```plain
> Table 1. AUROC comparison between layer-wise features and averaged multi-layer features in DADM.
> ```
>
> | opt-6.7b   | TruthfulQA | TriviaQA | CommonSenseQA | SciQ  |
> | ---------- | ---------- | -------- | ------------- | ----- |
> | Average    | 66.93      | 57.91    | 62.05         | 69.94 |
> | Layer-wise | 79.57      | 65.68    | 79.33         | 80.68 |
>
>
> **Q3.** On the Reliability of the Evaluation
>
> **A4:** We agree with the reviewer that BLEURT-based labels can introduce noise, which may affect the performance of DADM. We plan to explore alternative evaluation methods including the "LLM-as-a-judge" approach, to ensure that our method is robust to label noise and can maintain high reliability. We will report the corresponding results in the later stage of the rebuttal.
>
> [1] Du, Xuefeng; Xiao, Chaowei; Li, Yixuan. HaloScope: Harnessing Unlabeled LLM Generations for Hallucination Detection. NeurIPS 2024
> [2] Park, Seongheon; Du, Xuefeng; Yeh, Min-Hsuan; Wang, Haobo; Li, Yixuan. Steer LLM Latents for Hallucination Detection. ICML 2025

---

> > ### Comment · Reviewer_uhnD · 2025-11-25
> >
> > Thank you for the response. Some of my concerns have been addressed. But I still concern the robustness of the proposed method, so I keep my score.

---

> > > ### Author Response · Authors · 2025-12-04
> > > **Response to Reviewer uhnD (Follow up)**
> > >
> > > We sincerely thank the reviewer for the response. Here are our new findings concerning Q3.
> > >
> > > **Q3.** On the Reliability of the Evaluation
> > >
> > > **A4:** To ensure a more comprehensive and reliable evaluation, we further assess our method under two alternative metrics: ROUGE evaluation and the DeepSeek-V3.2 LLM-as-a-judge evaluation, which has been shown to align better with human preferences. These results in Table 1 validate that the effectiveness of DADM is not dependent on a specific metric, but generalizes well across diverse evaluation metrics.
> > >
> > > ```plain
> > > Table 1. Performance comparison of hallucination detection methods across OPT-6.7b with ROUGE and DeepSeek labeling.
> > > ```
> > >
> > > | **Metric**   | **Method**             | **TruthfulQA** | **TriviaQA** | **CommonSenseQA** | **SciQ**  |
> > > | ------------ | ---------------------- | -------------- | ------------ | ----------------- | --------- |
> > > | **ROUGE**    | **Perplexity**         | 55.99          | 50.23        | 60.94             | 51.29     |
> > > |              | **Fisher Rao**         | 52.43          | 51.38        | 57.61             | 59.92     |
> > > |              | **Lexical Similarity** | 50.16          | 51.38        | 68.39             | 51.08     |
> > > |              | **MSP**                | 53.08          | 52.76        | 52.71             | 50.29     |
> > > |              | **Semantic Entropy**   | 53.42          | 56.54        | 64.61             | 53.04     |
> > > |              | **EigenScore**         | 54.76          | 50.91        | 55.67             | 65.33     |
> > > |              | **HaloScope**          | 81.08          | 84.66        | **90.54**         | 80.24     |
> > > |              | **DADM (Ours)**        | **85.26**      | **86.11**    | 84.19             | **85.19** |
> > > | **DeepSeek** | **Perplexity**         | 53.38          | 50.67        | 52.30             | 54.83     |
> > > |              | **Fisher Rao**         | 56.32          | 51.25        | 56.98             | 54.62     |
> > > |              | **Lexical Similarity** | 55.72          | 51.77        | 50.92             | 50.58     |
> > > |              | **MSP**                | 58.91          | 52.15        | 56.04             | 57.40     |
> > > |              | **Semantic Entropy**   | 52.21          | 54.24        | 52.11             | 51.58     |
> > > |              | **EigenScore**         | 54.29          | 51.47        | 52.92             | 51.00     |
> > > |              | **HaloScope**          | 63.31          | 58.45        | 51.13             | 59.02     |
> > > |              | **DADM (Ours)**        | **67.62**      | **60.28**    | **63.76**         | **62.16** |

---

### Official Review · Reviewer_nnQd · 2025-10-29

**Soundness:** 2
**Presentation:** 3
**Contribution:** 2
**Rating:** 4
**Confidence:** 3

**Summary:**

The paper proposes Distance-Aware Distribution Modeling (DADM) for unsupervised LLM hallucination detection. DADM (1) purifies generations via an iterative, distance-based selector that retains consistently truthful samples and (2) fits a normalizing-flow model to the purified set, yielding likelihood-based confidence scores that maximize likelihood for truthful outputs and minimize it for hallucinations. This two-stage design captures global semantic/distributional structure while avoiding noise contamination from mixed-quality data, producing interpretable scores and reliable detection. On multiple benchmarks and LLM settings, DADM consistently outperforms prior unsupervised baselines.

**Strengths:**

1. The topic is interesting and addresses an important problem.
2. The paper is well written.
3. The paper is supported by solid theory.

**Weaknesses:**

1. The baseline model should add some new models, like LLaMA 3.2, Qwen2.5, Qwen 3.

2. For the benchmark discussion, note that several recent studies address both hallucination and maintain performance (even some improvement) on general scenario. I recommend the authors add some benchmarks like math etc.

**Questions:**

See above

---

> ### Author Response · Authors · 2025-11-21
> **Response to Reviewer nnQd**
>
> We thank Reviewer nnQd for the insightful comments. We will incorporate newer models and more diverse benchmarks to improve the overall scope and evaluation of our method.
>
> **W1**. The baseline model should add some new models, like LLaMA 3.2, Qwen2.5, Qwen 3.
> **A1**: Thank you for the suggestion. We agree that incorporating newer models such as Qwen3 can further strengthen our evaluation. We have now expanded our experiments to include these models to better demonstrate the broad applicability of DADM. The comparison results with HaloScope are presented here, and the results for the remaining baseline methods will be added to both the manuscript and the rebuttal as soon as they are ready.
>
> ```plain
> Table 1. AUROC (%) of normalizing flows on Qwen3 model.
> ```
>
> | qwen3-4b  | TruthfulQA | TriviaQA | CommonSenseQA | SciQ  |
> | --------- | ---------- | -------- | ------------- | ----- |
> | Haloscope | 72.81      | 61.46    | 60.66         | 61.81 |
> | DAMD      | 75.74      | 69.34    | 70.63         | 67.34 |
>
>
> **W2**. For the benchmark discussion, note that several recent studies address both hallucination and maintain performance (even some improvement) on general scenarios. I recommend the authors add some benchmarks like math, etc.
>
> **A2**: We appreciate the reviewer’s suggestion to include additional benchmarks. We note that [1] highlights the challenges of hallucination in mathematical reasoning and proposes a fine-grained step-level detection and mitigation framework. We would like to clarify how our work differs and where the current scope lies. Our method focuses on feature extraction from pre-trained LLM layers and subsequent distribution modeling, without fine-tuning the LLM. Because of this focus, our evaluation has so far centered on standard QA scenarios where feature distributions are more tractable. Although mathematical reasoning tasks may require specialized models tailored to reasoning and computational challenges, we are  planning additional experiments to extend our approach to such domains. We will update the manuscript once results become available. Thank you for the insightful suggestion!
>
> [1]Li, Ruosen; Luo, Ziming; Du, Xinya. FG-PRM: Fine-grained Hallucination Detection and Mitigation in Language Model Mathematical Reasoning. Findings of EMNLP 2025

---

> > ### Author Response · Authors · 2025-12-04
> > **Response to Reviewer nnQd (Follow up)**
> >
> > **W2**. For the benchmark discussion, note that several recent studies address both hallucination and maintain performance (even some improvement) on general scenarios. I recommend the authors add some benchmarks like math, etc.
> >
> > **A2**:  We thank the reviewer for the insightful suggestion. We believe our scope for this research is QA-based hallucination, while math style hallucination may be caused by a logical fault, which is out of our scope. Therefore, we add a long-form QA task on CLAPNQ dataset. We utilize Qwen3-4b as the language model and employ DeepSeek-V3.2 as the LLM-judge metric. DADM achieves competitive performance compared to baseline methods. It's noted that DADM achieves the highest score of 64.34, surpassing all other methods.
> >
> > ```plain
> > Table 1. AUROC (%) of different methods evaluated on LLaMA-2-7b using the TruthfulQA dataset.
> > ```
> >
> > | **Model**    | **Method**      | **CLAPNQ** |
> > | ------------ | --------------- | ---------- |
> > | **Qwen3-4b** | **Perplexity**  | 50.37      |
> > |              | **Fisher Rao**  | 57.93      |
> > |              | **HaloScope**   | 60.37      |
> > |              | **TSV**         | 62.78      |
> > |              | **DADM (Ours)** | 64.34      |

---

### Official Review · Reviewer_WeFf · 2025-11-01

**Soundness:** 2
**Presentation:** 3
**Contribution:** 2
**Rating:** 4
**Confidence:** 3

**Summary:**

This paper proposes a novel unsupervised framework called Distance-Aware Distribution Modeling (DADM) for detecting hallucinations in large language models.
DADM operates in two stages: In stage one, an iterative distance-based process is proposed to select a high-confidence subset of truthful samples using a covariance-modified Mahalanobis distance. This stage progressively filters out likely hallucinations, resulting in a coherent set of high-confidence responses. In stage two, a normalizing flow model is trained on the selected truthful subset to model the underlying distribution of truthful responses, enabling accurate likelihood estimation for hallucination detection.
The paper demonstrates that DADM consistently outperforms prior unsupervised methods across multiple datasets and LLMs. The key innovation lies in addressing the limitations of prior work (like HaloScope) that suffer from noise contamination in subspace estimation by first obtaining a clean subset of truthful responses.

**Strengths:**

1. The two-stage approach addresses a fundamental limitation of prior work. The key insight of first selecting a clean subset of truthful samples before modeling the distribution represents a creative solution to the noise contamination problem in existing subspace-based methods.
2. The findings of the paper are quite well supported by the experiments and the results are consistently strong across multiple datasets and LLMs, with statistically significant improvements over baselines.

**Weaknesses:**

1. Overly idealistic truthful subset assumption. The method presumes that truthful responses form a single, tight cluster. This assumption is unreasonable. For open-ended scenarios, multiple semantically distinct but equally correct answers form several clusters in the feature space. The proposed method might discard some small truthful clusters as the outliers. Moreover, consistent hallucinations might also form a tight cluster that gets selected as the truthful subset, which leads to the amplified mistake in the proposed second stage.
2. Hard label training with complement as negative is ill-posed. Treating the selected subset as positive hard labels and suppressing the complement as negative samples introduces substantial label noise.
3. Truthful sample ratio experiment lacks oracle baselines. Considering two ablation studies: (a) only use 0.1-0.7 ratio of BLEURT-true samples as the truthful subset for stage 2 modeling without complement treated as negative samples. (b) use BLEURT-true samples as the truthful subset with complement treated as negative samples. These two experiments would clearly disentangle the performance bottlenecks.

**Questions:**

1. How does your method handle scenarios where (a) truthful answers are multi-clustered, or (b) systematic hallucinations form their own dense clusters?
2. The Stage 2 objective treats the complement of the selected subset as negative samples, actively suppressing the likelihood. Since the complement inevitably contains many valid truthful responses, isn’t your model being explicitly trained to penalize the diversity of truthful answers?
3. Could you add the oracle ablations: (a) train stage 2 on only 0.1-0.7 fractions of BLEURT-true subset, (b) train stage 2 on 0.1-0.7 fractions of BLEURT-true subset and the complement as negative samples? This quantifies the upper bound and the impact of noisy labels.
4. Have you tried multi-layer feature aggregation or layer-invariant representations to reduce the sensitivity to layer choice?
5. The paper mentions the use of the softplus function in the "Hallucination Suppression" section. However, the softplus function is not explicitly utilized in the described framework. Could you clarify this?
6. The entire evaluation is based on the BLEURT scores, which are merely a proxy for human judgment and have their own biases. Recent work [1] conducted the “LLM-as-a-judge” evaluation, which is considered as a more reliable and human-aligned evaluation method. Could you add a additional ablation to verify the validity of your method?


[1] https://arxiv.org/abs/2503.01917

---

> ### Author Response · Authors · 2025-11-21
> **Response to Reviewer WeFf (1)**
>
> We sincerely thank the reviewer for the thoughtful comments and valuable suggestions. We address each of the points you raised in detail below and will incorporate these clarifications in the revised version of the manuscript.
>
> **W1 & Q1**. Overly idealistic truthful subset assumption & Handle different scenarios
>
> **A1**: We appreciate the reviewer's comment regarding the assumption that truthful answers form a tight cluster. To clarify, our approach does not assume that all truthful responses form a tight cluster. Instead, our method is designed to prioritize identifying a subset of truthful responses. In open-ended scenarios, there may indeed be multiple semantically valid answers that form several distinct clusters in the feature space. Our approach focuses on selecting the most confident cluster of truthful responses, prioritizing reliability to achieve more accurate hallucination detection.
>
> We hypothesize that systematic hallucinations tend to be more diverse and scattered in the feature space. This assumption is based on the fact that hallucinations are often unpredictable and can vary significantly in content and structure. However, if hallucinations cluster together in a particular region, their likelihood will increase and the likelihood of truthful responses will decrease in our method. Since we use a validation set to determine the correct sign of the scores, the model can adjust the scoring direction and still accurately distinguish truthful responses from hallucinations, even when hallucinations form a compact cluster in the feature space.
>
> **W2 & Q2**. Hard label training with complement as negative & Penalizing diversity of truthful answers
>
> **A2**: We understand the concern regarding treating the complement of the truthful subset as negative samples, which may introduce label noise by penalizing valid truthful responses. However, we chose this conservative approach to prioritize true positive detection, even at the cost of misclassifying a few truthful samples. We also observed in the ablation study (Fig 2) that when the ratio of truthful samples is relatively low (e.g., 0.1 to 0.3), the method consistently detects hallucinations effectively while still avoiding missed true positives.
>
> To further assess the impact of this conservative strategy, we conducted an ablation study where we only used the truthful subset to training normalizing flow model in Table 1 (without penalty). The results of this study show that by penalizing the complementary set, our method gains better performance.
>
> ```plain
> Table 1. AUROC (%) of normalizing flows without and with penalty on the complement set.
> ```
>
> | opt-6.7b        | TruthfulQA | TriviaQA | CommonSenseQA | SciQ  |
> | --------------- | ---------- | -------- | ------------- | ----- |
> | With Penalty    | 79.57      | 65.68    | 79.33         | 80.68 |
> | Without Penalty | 73.30      | 63.81    | 78.95         | 76.41 |
>
>
> **W3 & Q3**. Truthful sample ratio experiment lacks oracle baselines
>
> **A3**: We appreciate the suggestion to include oracle baselines for evaluating the truthful sample ratio. To address this, we conduct oracle experiments using BLEURT-true samples under two configurations: (a) using all BLEURT-true samples in the training set with penalization of the complement, and (b) using all BLEURT-true samples without applying the penalty. The results are summarized in Table 2. The oracle experiments reveal that using all BLEURT-true samples provides a clear performance upper bound for our method. Importantly, the default DADM configuration performs competitively relative to the oracle settings.
>
> ```plain
> Table 2. AUROC (%) of normalizing flow oracle baselines using BLEURT-true samples.
> ```
>
> | opt-6.7b                                  | TruthfulQA | TriviaQA | CommonSenseQA | SciQ  |
> | ----------------------------------------- | ---------- | -------- | ------------- | ----- |
> | Using BLEURT-true samples With Penalty    | 85.43      | 71.62    | 81.91         | 76.37 |
> | Using BLEURT-true samples Without Penalty | 80.15      | 63.92    | 79.63         | 75.14 |
> | DADM default                              | 79.57      | 65.68    | 79.33         | 80.68 |

---

> ### Author Response · Authors · 2025-11-21
> **Response to Reviewer WeFf (2)**
>
> **Q4**. Multi-layer feature aggregation or layer-invariant representations
>
> **A4**: In the current implementation of DADM, we mainly rely on features extracted from individual layers. Aggregating features across layers would produce extremely high-dimensional vectors (32 × 4096), which are not suitable for normalizing flows. Instead, we experimented with averaging features across all layers as a more tractable alternative. However, this aggregated representation shows noticeably worse performance than using features from a specific layer. The corresponding results are reported in Table 3.
>
> ```plain
> Table 3. AUROC comparison between layer-wise features and averaged multi-layer features in DADM.
> ```
>
> | opt-6.7b   | TruthfulQA | TriviaQA | CommonSenseQA | SciQ  |
> | ---------- | ---------- | -------- | ------------- | ----- |
> | Average    | 66.93      | 57.91    | 62.05         | 69.94 |
> | Layer-wise | 79.57      | 65.68    | 79.33         | 80.68 |
>
>
> **Q5**. Clarification on the use of the softplus function
>
> **A5**: We apologize for the confusion caused by the mention of the softplus function.  In our implementation, the softplus function is used in the hallucination-suppression term  to avoid numerical instability when the likelihood becomes extremely small.  Directly penalizing $ \log p_\mathcal{F}(f_j) $ can lead to issues such as  $ \log p_\mathcal{F}(f_j) \to -\infty $, which causes gradient explosion and unstable training.  To address this, we replace the raw log-likelihood with its softplus transformation:  $ \text{Softplus}(\log p_\mathcal{F}(f_j))
> = \log\big(1 + \exp(\log p_\mathcal{F}(f_j))\big)
> = \log(1 + p_\mathcal{F}(f_j)) \to 0. $  This transformation  ensures that the loss remains finite even for extremely small likelihood values, thereby stabilizing optimization while still pushing  hallucinated samples away from the truthful distribution.  We will update the manuscript to clarify this design choice.
>
> **Q6**. Evaluation based on BLEURT scores
>
> **A6**: We appreciate the reviewer for highlighting the potential biases of using BLEURT scores as proxies for human judgment. As suggested, we are currently conducting additional evaluations using an “LLM-as-a-judge” hallucination assessment based on a stronger LLM. We will update the rebuttal and manuscript once these results are available.

---

> > ### Author Response · Authors · 2025-12-04
> > **Response to Reviewer WeFf (Follow up)**
> >
> > **Q6**. Evaluation based on BLEURT scores
> >
> > **A6**: To ensure a more comprehensive and reliable evaluation, we now further assess our method under two alternative metrics: ROUGE evaluation and the DeepSeek-V3.2 LLM-as-a-judge evaluation, which has been shown to align better with human preferences. These results in Table 1 validate that the effectiveness of DADM is not dependent on a specific metric, but generalizes well across diverse evaluation metrics.
> >
> > ```plain
> > Table 1. Performance comparison of hallucination detection methods across OPT-6.7b with ROUGE and DeepSeek labeling.
> > ```
> >
> > | **Metric**   | **Method**             | **TruthfulQA** | **TriviaQA** | **CommonSenseQA** | **SciQ**  |
> > | ------------ | ---------------------- | -------------- | ------------ | ----------------- | --------- |
> > | **ROUGE**    | **Perplexity**         | 55.99          | 50.23        | 60.94             | 51.29     |
> > |              | **Fisher Rao**         | 52.43          | 51.38        | 57.61             | 59.92     |
> > |              | **Lexical Similarity** | 50.16          | 51.38        | 68.39             | 51.08     |
> > |              | **MSP**                | 53.08          | 52.76        | 52.71             | 50.29     |
> > |              | **Semantic Entropy**   | 53.42          | 56.54        | 64.61             | 53.04     |
> > |              | **EigenScore**         | 54.76          | 50.91        | 55.67             | 65.33     |
> > |              | **HaloScope**          | 81.08          | 84.66        | **90.54**         | 80.24     |
> > |              | **DADM (Ours)**        | **85.26**      | **86.11**    | 84.19             | **85.19** |
> > | **DeepSeek** | **Perplexity**         | 53.38          | 50.67        | 52.30             | 54.83     |
> > |              | **Fisher Rao**         | 56.32          | 51.25        | 56.98             | 54.62     |
> > |              | **Lexical Similarity** | 55.72          | 51.77        | 50.92             | 50.58     |
> > |              | **MSP**                | 58.91          | 52.15        | 56.04             | 57.40     |
> > |              | **Semantic Entropy**   | 52.21          | 54.24        | 52.11             | 51.58     |
> > |              | **EigenScore**         | 54.29          | 51.47        | 52.92             | 51.00     |
> > |              | **HaloScope**          | 63.31          | 58.45        | 51.13             | 59.02     |
> > |              | **DADM (Ours)**        | **67.62**      | **60.28**    | **63.76**         | **62.16** |

---

### Author Response · Authors · 2025-12-04
**Summary(1/3)**

We sincerely appreciate the valuable feedback from the reviewers. Based on the suggestions, we have incorporated additional experiments to ensure a more comprehensive evaluation of our approach. Below is the detailed summary of the additional experiments we conducted,  and all these results have been included in the revised version of the manuscript:

+ **TSV Baseline**: To ensure a more comprehensive comparison, we further include the TSV for LLaMA-3.1-8b and Qwen3-4b in Table 1 and Table 2, our DADM method consistently surpasses or matches TSV, achieving notable gains on CommonSenseQA and SciQ, which highlights the robustness and general superiority of our approach across diverse evaluation scenarios.

```plain
Table 1.  Performance comparison of hallucination detection methods across LLaMA-3.1-8b model on four datasets.
```

| **Model**    | **Method**      | **TruthfulQA** | **TriviaQA** | **CommonSenseQA** | **SciQ**  |
| ------------ | --------------- | -------------- | ------------ | ----------------- | --------- |
| LLaMA-3.1-8b | **HaloScope**   | 70.27          | 73.51        | 64.66             | 76.01     |
|              | **TSV**         | **71.78**      | **78.59**    | 67.83             | 70.40     |
|              | **DADM (Ours)** | 71.58          | 77.06        | **78.30**         | **80.24** |


```plain
Table 2.  Performance comparison of hallucination detection methods across Qwen3-4b model on four datasets.
```

| **Model**    | **Method**      | **TruthfulQA** | **TriviaQA** | **CommonSenseQA** | **SciQ**  |
| ------------ | --------------- | -------------- | ------------ | ----------------- | --------- |
| LLaMA-3.1-8b | **HaloScope**   | 72.81          | 61.46        | 60.66             | 61.81     |
|              | **TSV**         | 64.64          | 67.14        | 68.47             | 66.96     |
|              | **DADM (Ours)** | **75.74**      | **69.34**    | **70.63**         | **67.34** |


+ **Oracle Baselines**: We conduct oracle experiments using BLEURT-true samples under two configurations: (a) using all BLEURT-true samples in the training set with penalization of the complement, and (b) using all BLEURT-true samples without applying the penalty. The results are summarized in Table 3. The oracle experiments reveal that using all BLEURT-true samples provides a clear performance upper bound for our method. Importantly, the default DADM configuration performs competitively relative to the oracle settings.

```plain
Table 3. AUROC (%) of normalizing flow oracle baselines using BLEURT-true samples.
```

| opt-6.7b                                  | TruthfulQA | TriviaQA | CommonSenseQA | SciQ  |
| ----------------------------------------- | ---------- | -------- | ------------- | ----- |
| Using BLEURT-true samples With Penalty    | 85.43      | 71.62    | 81.91         | 76.37 |
| Using BLEURT-true samples Without Penalty | 80.15      | 63.92    | 79.63         | 75.14 |
| DADM default                              | 79.57      | 65.68    | 79.33         | 80.68 |

---

> ### Author Response · Authors · 2025-12-04
> **Summary(2/3)**
>
> + **ROUGE and LLM Evaluation**: To ensure a more comprehensive and reliable evaluation, we further assess our method under two alternative metrics: ROUGE evaluation and the DeepSeek-V3.2 LLM-as-a-judge evaluation, which has been shown to align better with human preferences. These results in Table 4 validate that the effectiveness of DADM is not dependent on a specific metric, but generalizes well across diverse evaluation metrics.
>
> ```plain
> Table 4. Performance comparison of hallucination detection methods across OPT-6.7b with ROUGE and DeepSeek labeling.
> ```
>
> | **Metric**   | **Method**             | **TruthfulQA** | **TriviaQA** | **CommonSenseQA** | **SciQ**  |
> | ------------ | ---------------------- | -------------- | ------------ | ----------------- | --------- |
> | **ROUGE**    | **Perplexity**         | 55.99          | 50.23        | 60.94             | 51.29     |
> |              | **Fisher Rao**         | 52.43          | 51.38        | 57.61             | 59.92     |
> |              | **Lexical Similarity** | 50.16          | 51.38        | 68.39             | 51.08     |
> |              | **MSP**                | 53.08          | 52.76        | 52.71             | 50.29     |
> |              | **Semantic Entropy**   | 53.42          | 56.54        | 64.61             | 53.04     |
> |              | **EigenScore**         | 54.76          | 50.91        | 55.67             | 65.33     |
> |              | **HaloScope**          | 81.08          | 84.66        | **90.54**         | 80.24     |
> |              | **DADM (Ours)**        | **85.26**      | **86.11**    | 84.19             | **85.19** |
> | **DeepSeek** | **Perplexity**         | 53.38          | 50.67        | 52.30             | 54.83     |
> |              | **Fisher Rao**         | 56.32          | 51.25        | 56.98             | 54.62     |
> |              | **Lexical Similarity** | 55.72          | 51.77        | 50.92             | 50.58     |
> |              | **MSP**                | 58.91          | 52.15        | 56.04             | 57.40     |
> |              | **Semantic Entropy**   | 52.21          | 54.24        | 52.11             | 51.58     |
> |              | **EigenScore**         | 54.29          | 51.47        | 52.92             | 51.00     |
> |              | **HaloScope**          | 63.31          | 58.45        | 51.13             | 59.02     |
> |              | **DADM (Ours)**        | **67.62**      | **60.28**    | **63.76**         | **62.16** |
>
>
> + **Effect of Penalty**: To further assess the impact of this conservative strategy,  we conducted an ablation study where we only used the truthful subset to train the normalizing flow model in Table 5 (without penalty). The results of this study show that penalizing the complementary set improves our method's performance.
>
> ```plain
> Table 5. AUROC (%) of normalizing flows without and with penalty on the complement set.
> ```
>
> | opt-6.7b        | TruthfulQA | TriviaQA | CommonSenseQA | SciQ  |
> | --------------- | ---------- | -------- | ------------- | ----- |
> | With Penalty    | 79.57      | 65.68    | 79.33         | 80.68 |
> | Without Penalty | 73.30      | 63.81    | 78.95         | 76.41 |

---

> > ### Author Response · Authors · 2025-12-04
> > **Summary(3/3)**
> >
> > + **Averaged Multi-layer Features**:  We experimented with averaging features across all layers as a more tractable alternative. However, this aggregated representation shows noticeably worse performance than using features from a specific layer. The corresponding results are reported in Table 6.
> >
> > ```plain
> > Table 6. AUROC comparison between layer-wise features and averaged multi-layer features in DADM.
> > ```
> >
> > | opt-6.7b   | TruthfulQA | TriviaQA | CommonSenseQA | SciQ  |
> > | ---------- | ---------- | -------- | ------------- | ----- |
> > | Average    | 66.93      | 57.91    | 62.05         | 69.94 |
> > | Layer-wise | 79.57      | 65.68    | 79.33         | 80.68 |
> >
> >
> > + **Generalization**: We now present the cross-dataset AUROC performance across the four QA datasets in Table 7 to demonstrate the generalizability. Models trained on one dataset consistently maintain competitive performance when transferred to the remaining datasets.
> >
> > ```plain
> > Table 7. Cross-dataset generalizability results across four QA datasets.
> > ```
> >
> > | Source \ Target | **TruthfulQA** | **TriviaQA** | **Commonsense** | **SciQ** |
> > | --------------- | -------------- | ------------ | --------------- | -------- |
> > | **TruthfulQA**  | 79.57          | 65.17        | 79.81           | 69.62    |
> > | **TriviaQA**    | 69.87          | 65.68        | 72.57           | 73.25    |
> > | **Commonsense** | 75.65          | 64.14        | 79.93           | 75.43    |
> > | **SciQ**        | 64.66          | 64.65        | 78.28           | 80.68    |
> >
> >
> > + **Long-form QA results**: We now add a long-form QA task on CLAPNQ dataset. We utilize Qwen3-4b as the language model and employ DeepSeek-V3.2 as the LLM-judge metric. DADM achieves superior performance compared to baseline methods. Note that DADM yields the highest score of 64.34, surpassing all other methods.
> >
> > ```plain
> > Table 8. AUROC (%) of different methods evaluated on LLaMA-2-7b using the TruthfulQA dataset.
> > ```
> >
> > | **Model**    | **Method**      | **CLAPNQ** |
> > | ------------ | --------------- | ---------- |
> > | **Qwen3-4b** | **Perplexity**  | 50.37      |
> > |              | **Fisher Rao**  | 57.93      |
> > |              | **HaloScope**   | 60.37      |
> > |              | **TSV**         | 62.78      |
> > |              | **DADM (Ours)** | 64.34      |

---

### Meta-Review · Area_Chair_vNqg · 2025-12-28

**Summary:**

The paper proposes DADM, a two-stage unsupervised framework for LLM hallucination detection. In stage one, an iterative distance-based process is proposed to select a high-confidence subset of truthful samples using a covariance-modified Mahalanobis distance. In stage two, a normalizing flow model is trained on the selected truthful subset to model the underlying distribution of truthful responses, enabling accurate likelihood estimation for hallucination detection.
While clearly presented, its experiments and technical validity are considered insufficient to justify acceptance at this time.
Considering the reviewers’ concerns, we regret that the paper cannot be recommended for acceptance at this time. The authors are encouraged to consider the reviewers’ comments when revising the paper for submission elsewhere.

**Reviewer Concerns:**

Key concerns include (1) Overly idealistic truthful subset assumption, (2) Hard label training with complement as negative is ill-posed, (2) Hard label training with complement as negative is ill-posed, (3) High hyperparameter sensitivity, (4) generalize to out-of-distribution data.

**Reviewer Scores:**

Across reviewers, soundness and contribution are consistently rated “fair,” with good presentation. All ratings cluster at 4 (marginally below acceptance), indicating recognition of results but insufficient robustness and methodological clarity to meet acceptance standards.

---

### Decision · Program_Chairs · 2026-01-26

Reject